# Influence of gas migration on permeability of soft coalbed methane reservoirs under true triaxial stress conditions

Gang Wang[1,2], Zhiyuan Liu[2], Yanwei Hu[2], Cheng Fan[2], Wenrui Wang[2] and Jinzhou Li[2]

[1]Mine Disaster Prevention and Control-Ministry of State Key Laboratory Breeding Base, and [2]College of Mining and Safety Engineering, Shandong University of Science and Technology, Qingdao 266590, People's Republic of China

 GW, 0000-0003-4742-8103

energy/engineering geology/environmental engineering

**Keywords:**
true triaxial stress, soft coalbed, gas migration, permeability model

**Author for correspondence:**
Gang Wang
e-mail: gang.wang@sdust.edu.cn

The permeability of the coal body is the key parameter restricting the efficient extraction of coalbed methane, and scholars have analysed it from two angles of the change of stress state and porosity of the coal body. However, there is still a lack of study on the mechanism of gas migration and movement in soft coalbed methane reservoir under the coupling between the true triaxial stress field (maximum principal stress $\sigma_1$ > intermediate principal stress $\sigma_2$ > minimum principal stress $\sigma_3$) and the gas pressure field. In this paper, the coal gas adsorption and seepage experiments are conducted through the self-developed true triaxial 'gas–solid' coupled coal mass seepage system with gas as the adsorption and seepage medium and coal briquette taking the place of soft coalbed methane reservoirs. Furthermore, the coal gas adsorption deformation model and the permeability evolution model taking gas adsorption into account are developed. Through analysis of both experimental and theoretic results, the main conclusions are drawn as follows: (i) With the increase in gas pressure, the adsorption deformation variation of coal mass is divided into a slow growth zone, a stable growth zone and a rapid growth zone. (ii) The gas adsorption deformation model developed can predict the variation trend of coal mass adsorption volumetric strains for different types of soft coalbeds, and the fitting variance of experimental and theoretical volumetric strains is above 98%. (iii) With the increase in maximum principal stress difference, the coal permeability variation curve shows two obvious turning points, which can be divided into a slow reduction zone, a rapid reduction zone and a steady reduction zone. (iv) The permeability model of coal mass considering the gas adsorption effect can reflect the variation characteristics of

permeability in the rapid reduction zone, and the overall fitting variance of experimental and theoretical permeabilities is above 91%. The above results could provide a reliable experimental and theoretical basis for improving coalbed methane extraction rates.

# 1. Introduction

During the coal seam mining process, coal mass in front of working faces is usually affected by mining disturbances and thus subject to unequal stresses in three directions. To be specific, the support pressure in the vertical direction increases and the pressure in the horizontal directions is relieved. As a consequence, expansion deformation of coal mass occurs [1], and further continuous development, expansion and penetration of pores and fracture structures in coal mass occur, finally leading to desorption, permeability enhancement and migration of gas in pores [2]. Especially in the presence of soft coalbeds, pore fracture development and gas migration become more severe, which have a dynamic influence on the stable extraction of coalbed methane. Therefore, understanding the mechanism of deformation damage and permeability evolution of soft coal mass under the coupling of gas and stress is essential to increase coalbed methane extraction rates [3] and realize scientific coal mining [4].

Many researchers have studied the adsorption of coalbed methane and the permeability of coal and rock mass through physical experiments. Meng & Li [5] and Connell et al. [6] studied the effect of gas adsorption on coal matrix and cleat deformation. In addition, some researchers focused on the permeability characteristics of coal mass and conducted a large number of physical experiments [7–11]. A few researchers investigated the seepage regulation of shale and sandstone under true triaxial stress conditions [12,13]. However, most of them put their main efforts in the experimental analysis of mechanical properties of coal under quasi-triaxial stresses, which could not reflect the actual stress state of coalbed methane reservoirs. Moreover, there is a lack of deep research on the mechanism behind the experimental phenomena.

In order to provide theoretical support for physical experiments, Palmer & Mansoori [14] examined the relationship between permeability and porosity of coalbed methane reservoirs under simple stress conditions. Shi & Durucan [15] developed a dynamic evolution model for the permeability of coalbed methane reservoirs taking into account the influence of effective stresses on the adsorption deformation and permeability of coal mass. Liu et al. [16] studied the effect of coalbed methane adsorption strain on coal permeability. Zang & Wang [17] proposed a model based on quasi-steady-state diffusion to reflect the relationship between gas adsorption deformation and permeability evolution [18].

A number of investigators explored the impact of coalbed methane adsorption on coal deformation and permeability under conventional triaxial stress conditions by combining physical experiments with theoretical derivation. Peng et al. [19] quantitatively described the degree of influence of coal pore expansion deformation and matrix deformation on the permeability of coal mass by means of the strain splitting function. Wei et al. [20] and Peng et al. [21] built a bi-directional permeability model considering the relationship between effective stresses and pore pressure. Liu et al. [22] introduced an internal expansion coefficient ($f$) to quantify the effect of coal mass matrix deformation caused by gas adsorption on coal pore diameter and permeability. Lu & Connell [23], Wang et al. [24] and Connell [25] improved the effective stress principle reflecting the influence of coal mass deformation, gas diffusion in coal matrix and gas migration in fractures on coal mass permeability with pore pressure [26], matrix pressure and gas adsorption pressure considered for the double porosity characteristics of coalbed methane reservoirs. Saurabh & Harpalani [27] suggested a coal permeability model of coal mass suitable for elastic and inelastic deformation of coal mass, but only the effect of stress was considered. Lu et al. [28] explored the effect of effective stresses and matrix adsorption deformation on coal permeability and developed a coal mass permeability model under specific boundary conditions.

Researchers above have studied the influence of coalbed methane adsorption deformation and coal porosity on coal permeability. However, the following insufficiencies exist: (i) owing to the limitations of test equipment, it is impossible to fully reflect the heterogeneity of permeability of coalbed methane reservoirs under real stress conditions, especially for soft coalbeds; (ii) the theoretical models of coal permeability developed have not considered coalbed methane adsorption.

The soft coal seam is the key reservoir to prevent coal and gas outburst and improve the pumping rate of coalbed methane because coal and gas outburst accidents are liable to occur in the soft coal seam where the coal is soft and brittle. Therefore, this paper is focused on the mechanical properties and seepage characteristics of soft coalbeds, which are carried out through the self-developed true triaxial

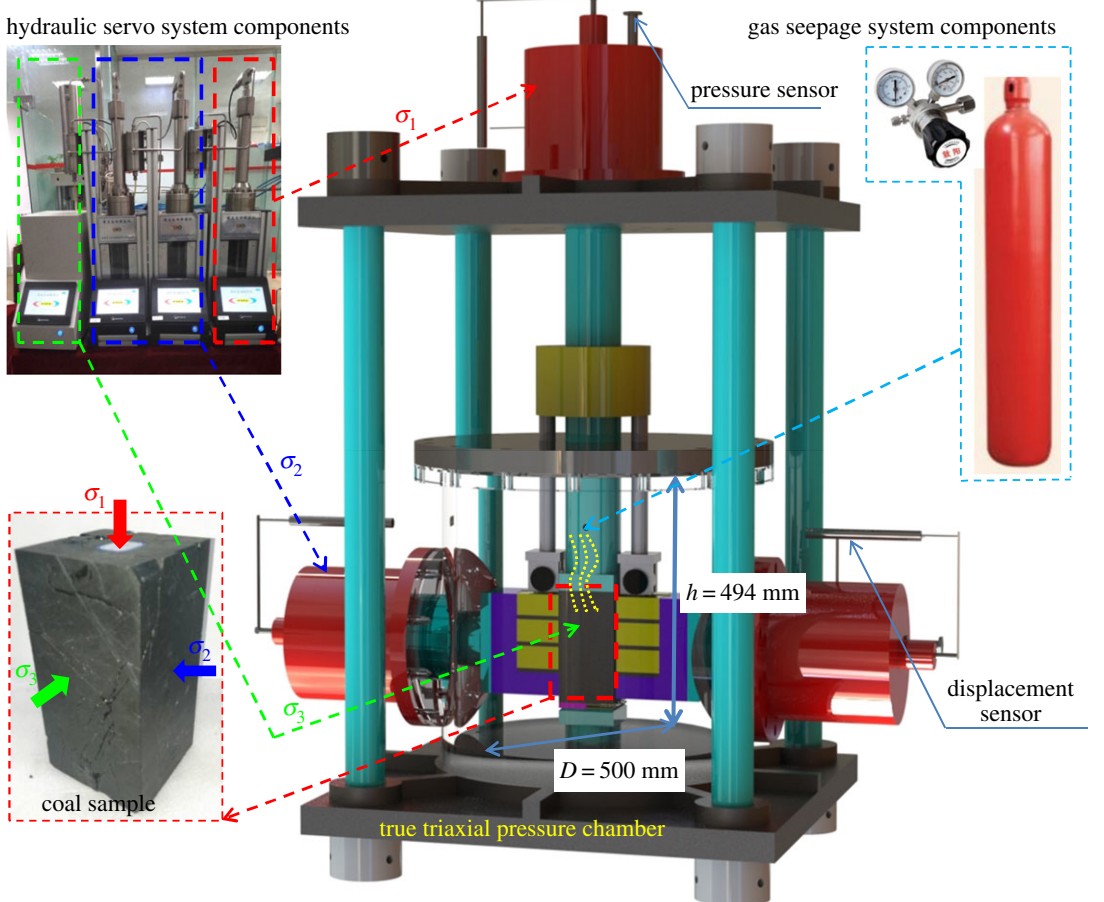

**Figure 1.** True triaxial gas–solid coupled coal mass seepage experimental system.

'gas–solid' coupled coal mass seepage system with gas as the adsorption and seepage medium and coal briquette taking the place of soft coalbed methane reservoirs. Moreover, the coal permeability continuity model considering the adsorption of coalbed methane is proposed. In addition, the deformation and permeability characteristics exhibited in the experiment and the theoretical model are compared and analysed, and the mechanism of gas migration in soft coalbed methane reservoirs is further understood. This could provide a theoretical support and experimental basis for efficient coalbed methane extraction.

## 2. Adsorption and seepage experiment for soft coal mass under true triaxial stress conditions

### 2.1. Experimental system

The self-developed true three-axis 'gas–solid' coupled coal mass seepage experimental system is used for this experiment [29]. As shown in figure 1, it is divided into four parts: a true triaxial pressure chamber, a hydraulic servo system, a gas seepage system and a data monitoring and control system. The true triaxial pressure chamber is the place for the gas adsorption and seepage experiment. The hydraulic servo system is the power source of true triaxial stresses. The gas seepage system is mainly composed of a pressure relief valve and a gas cylinder. The data monitoring system has a variety of high-precision sensors and data acquisition devices, which can monitor and collect the instantaneous deformation and gas flow information in real time. This apparatus has a feature of 'two rigid and one soft, tri-directional independent loading', which means that $\sigma_1$ and $\sigma_2$ of coal mass are rigid stresses and $\sigma_3$ is a flexible stress (owing to hydraulic oil). Therefore, this apparatus can be used to conduct gas adsorption and seepage experiments under a variety of complex loading and unloading stress conditions.

**Table 1.** Basic parameters of four kinds of coal samples.

| basic parameter | symbol and unit | Coal Sample 1 | Coal Sample 2 | Coal Sample 3 | Coal Sample 4 |
|---|---|---|---|---|---|
| pore bulk modulus | $K_p$ (GMPa) | 31.45 | 26.50 | 17.10 | 31.50 |
| initial porosity | $\phi$ (%) | 3.7 | 4.6 | 4.1 | 5.0 |
| bulk modulus of porous medium | $K_s$ (GMPa) | 850 | 570 | 420 | 630 |
| Young's modulus | $E_s$ (GMPa) | 990 | 760 | 400 | 680 |
| Poisson's ratio | $v_s$ | 0.30 | 0.28 | 0.26 | 0.32 |
| density | $\rho_s$ (g cm$^{-3}$) | 1.25 | 1.22 | 1.24 | 1.21 |
| pore diameter to diameter ratio | $c$ | 0.1 | | | |
| Langmuir volume | $V_L$ (cm$^3$ g$^{-1}$) | 17.7 (CH$_4$) | | | |
| Langmuir pressure | $P_L$ (MPa) | 7.2 (CH$_4$) | | | |
| volumetric strain coefficient associated with gas adsorption | $\varepsilon_g$ (g cm$^{-3}$) | $7.4 \times 10^{-4}$ | | | |

## 2.2. Preparation of soft coal samples

The raw coal for this experiment is taken from the 8# anthracite coal seam of Yuyang Coal Mine of Chongqing Songzao Coal and Electricity Co., Ltd. This coal seam has soft and brittle coal, and many coal and gas outburst accidents happened in working faces at different mining depths there. The specific industrial indicators for the 8# coal seam are as follows: volatile content 9.87–10.97%, ash content 11.53–19.13%, water content 0.56–2.55%, true density 1.5–1.53 g cm$^{-3}$, apparent density 1.34–1.38 g cm$^{-3}$, firmness coefficient 0.21–0.38, uniaxial compressive strength 0.89 MPa and coal mass failure type Class III–V [30].

The 8# anthracite coal seam is a soft coal seam. According to the experimental requirements for the sample size (100 mm × 100 mm × 200 mm), the raw coal is not suitable for this experiment because it is difficult to form and the raw coal pore fractures are randomly distributed. However, coal briquette has a similar variation trend in mechanical properties and permeability characteristics to that of raw coal [31], and it has good homogeneity suitable for repetitive experiments [7]. Therefore, the briquette samples are used for this experiment. According to 'GB/T 23561.9-2009 Methods for determining the physical and mechanical properties of coal and rock', the coal mass is processed into rectangular soft coal samples of 100 mm × 100 mm × 200 mm. After the basic mechanical test on these coal samples, four kinds of soft coal samples are selected for the gas adsorption deformation and permeability experiment with their basic parameters shown in table 1.

## 2.3. Experimental method and procedure

Two types of experiments are conducted: a constant stress experiment and a constant gas adsorption pressure experiment. In the gas adsorption deformation experiment, the external stress is kept constant, and in the coal mass seepage experiment, the gas pressure is kept constant. For the experimental procedure, the experiment under an initial true triaxial stress condition of 4 MPa for $\sigma_3$, 6 MPa for $\sigma_2$ and 8 MPa for $\sigma_1$ and an initial adsorption pressure of 1 MPa for $P$ is introduced as an example, as shown in figure 2.

Before the experiment begins, the overall system air tightness check is performed. When the reading of the gas flow accumulator is unchanged and the relation curve between gas flow and time tends to be horizontal, it is indicated that the air tightness is good. Now the experiment can be carried out.

(1) Gas adsorption experiment of coal mass under true triaxial stress conditions: First, increase $\sigma_1$, $\sigma_2$ and $\sigma_3$ gradually to a predetermined hydrostatic pressure of 4 MPa at a speed of 0.01 MPa s$^{-1}$. Next, keep $\sigma_3$ constant and increase $\sigma_1$ and $\sigma_2$ to 6 MPa. Then, increase $\sigma_1$ to 8 MPa. Now, a true triaxial stress environment is formed. After the stress environment is stable, open the high-pressure cylinder valve to adjust the inlet pressure to 1 MPa, and then close the inlet and outlet valves of the gas pipeline. Finally, carry out the gas adsorption deformation experiment. When the adsorption deformation

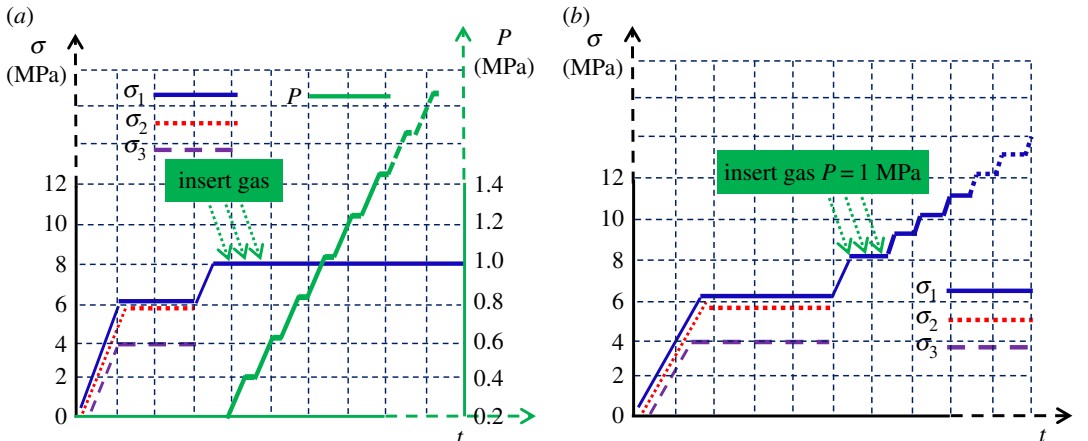

**Figure 2.** Schematic diagram of the experimental method and procedure: (*a*) gas adsorption path for coal samples and (*b*) permeability evolution path for coal samples.

curve of the coal sample tends to be horizontal, which indicates that the coal sample adsorption deformation is saturated, the amount of coal sample adsorption deformation is continued to be measured with the increase in the coal sample adsorption pressure at an interval of 0.2 MPa.

(2) Gas seepage experiment of coal mass under true triaxial stress conditions: After the initial true triaxial stress environment of 4 MPa for $\sigma_3$, 6 MPa for $\sigma_2$ and 8 MPa for $\sigma_1$ is formed based on the above steps, open the high-pressure cylinder valve slowly to adjust the inlet pressure to 1 MPa, so that the gas can enter the coal sample slowly. After the relation curve between gas flow and time tends to be horizontal, which means that the seepage is in the stable state, increase $\sigma_1$ gradually from 8 MPa at an interval of 1 MPa until the end of the experiment, during which the gas pressure is kept constant.

# 3. Test results and analysis

## 3.1. Effect of gas adsorption on the deformation of soft coal mass

Figure 3 shows the forms of four kinds of coal samples after the gas adsorption experiments. From the appearance of the coal samples, no obvious difference in their deformation forms is found. Figures 4–7 show the deformation curves of the coal samples caused by gas adsorption and measured under true triaxial stress conditions. When the external stress is kept constant, the gas adsorption pressure increases from 0 to 8 MPa in increments of 0.2 MPa. As the gas adsorption pressure increases, the strains ($\varepsilon_1$, $\varepsilon_2$, $\varepsilon_3$) of the four kinds of coal samples in the $\sigma_1$, $\sigma_2$ and $\sigma_3$ directions increase gradually and the volumetric strain ($\varepsilon_v$) also increases gradually. However, for each kind of coal sample, $\varepsilon_v$ is the maximum, followed by $\varepsilon_1$ and then $\varepsilon_3$, and $\varepsilon_2$ is the minimum, i.e. $\varepsilon_1 > \varepsilon_3 > \varepsilon_2$.

From figures 4–7, it can be seen that the deformation variation of coal mass exhibits different deformation rates under different gas adsorption pressures, and that it can be divided into a slow growth zone, a stable growth zone and a rapid growth zone. For the slow growth zone, when the gas adsorption pressure is lower than about 1 MPa, the strains in all three directions and volumetric strains of the coal samples change slightly. For example, the volumetric strain increases by 0.00084 for Coal Sample 1, by 0.00128 for Coal Sample 2, by 0.00079 for Coal Sample 3 and by 0.00198 for Coal Sample 4 (shown in table 2). The strains in the three directions are different. For the stable growth zone, when the gas pressure reaches about 5 MPa, the deformation of the coal samples is relatively stable, and the deformations in all three directions and volumetric strains show significant differences. For the rapid growth zone, when the gas pressure reaches the maximum value, the deformation in all three directions and volumetric strains of the coal samples are in a rapid growth stage.

From a microscopic point of view, the coal samples undergo expansion deformation after adsorbing gas. When the gas adsorption occurs, the gas molecules adhere to the surface of coal particles, resulting in a decrease in the surface tension of the coal particles. This means the attraction between the molecules on the surface of coal particles and the molecules inside is reduced and thus, the distance between them is increased. Therefore, the coal mass after its gas adsorption is more likely to be deformed than before [32].

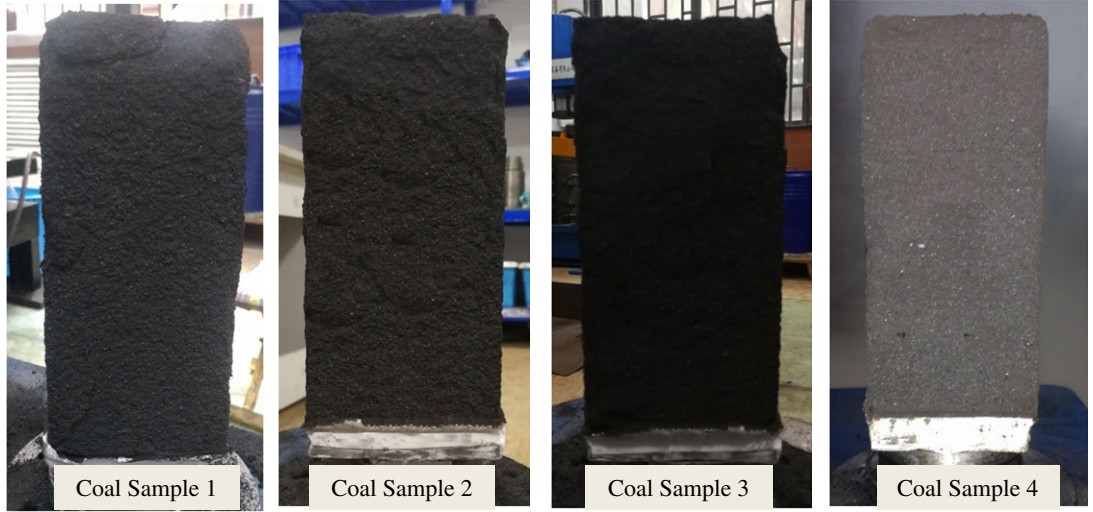

**Figure 3.** Forms of coal samples after gas adsorption experiments.

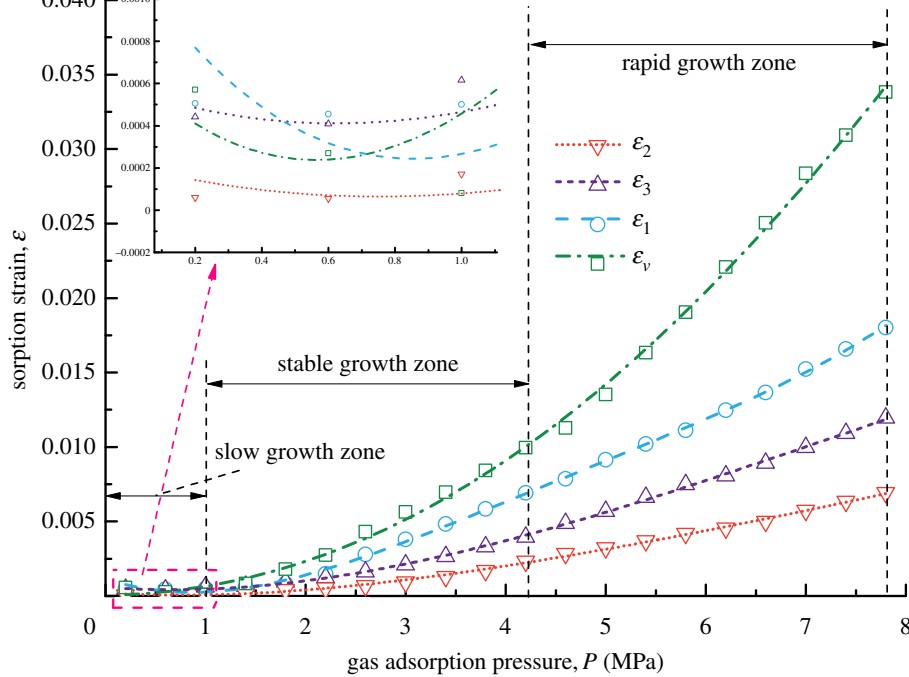

**Figure 4.** Deformation characteristics of Coal Sample 1 under different gas adsorption pressures.

From a macroscopic point of view, the coal samples in this experiment are cuboids, and their length in the $\sigma_1$ direction is twice those in the other two directions. Consequently, the adsorption deformation mainly occurs in the $\sigma_1$ direction. For the other two directions, the load controlled method is adopted in the $\sigma_2$ direction with a large stress, and the strain-controlled method is adopted in the $\sigma_3$ direction with a small stress. This indicates that the external binding force in the $\sigma_2$ direction is larger than that in the $\sigma_3$ direction, and thus deformation is prone to occur in the $\sigma_3$ direction. Therefore, the gas adsorption deformation exhibits the characteristics of $\varepsilon_1 > \varepsilon_3 > \varepsilon_2$.

In this gas adsorption experiment, gas pressure is the main factor of coal mass deformation. Gas pressure acts as both adsorption pressure and pore pressure in coal sample pore fractures. Specifically, the effect of gas adsorption plays a leading role under low gas pressure conditions, while the effect of pore pressure plays a dominant role under high gas pressure conditions.

When the external stress of the coal mass is constant and the gas adsorption pressure is less than 5 MPa, the volumetric strains of the coal samples depend on the gas adsorption of the coal samples. Moreover, gas adsorption is related to gas pressure. As gas pressure increases, gas adsorption becomes stronger, and more

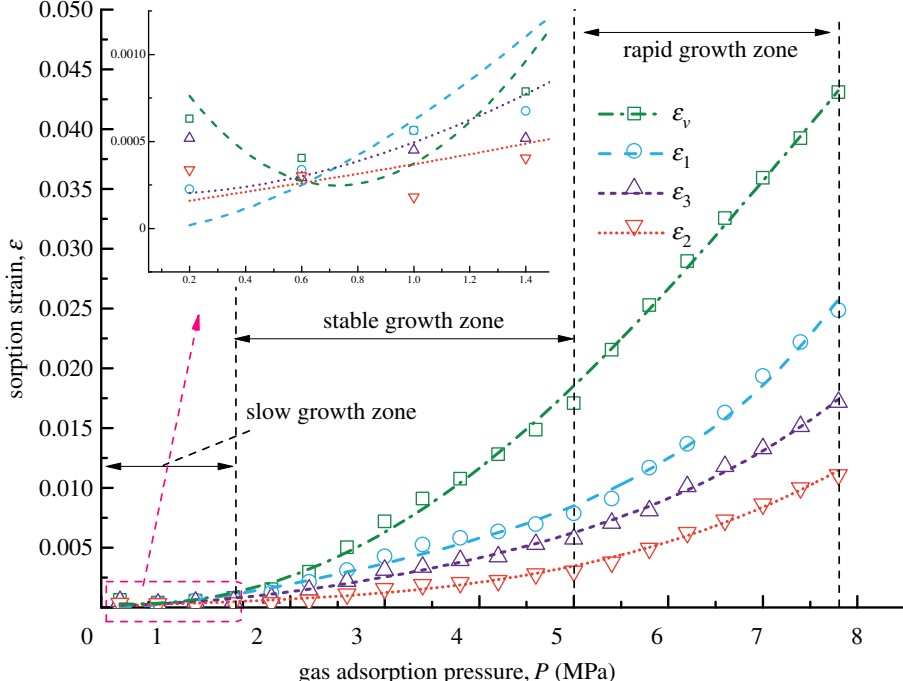

**Figure 5.** Deformation characteristics of Coal Sample 2 under different gas adsorption pressures.

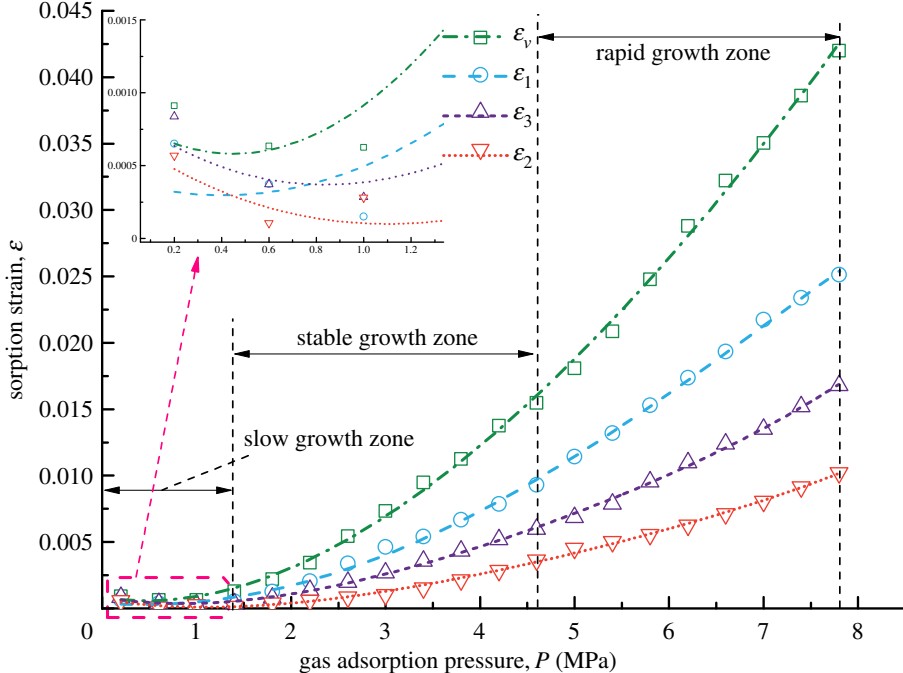

**Figure 6.** Deformation characteristics of Coal Sample 3 under different gas adsorption pressures.

gas is adsorbed on the pore surface of the coal samples, resulting in a larger adsorption thickness. Consequently, the transfer resistance between gas molecules increases. When the gas pressure is low, the pore pressure is small, and thus, it is difficult to cause the deformation of coal mass. Therefore, the volumetric strains of the coal samples in this zone are mainly determined by gas adsorption.

When the gas adsorption pressure is greater than 5 MPa, the gas pressure exhibits the effect of pore pressure other than the effect of adsorption, and the interiors of the coal samples expand outward owing to pore pressure. This indicates that the volumetric strains of the coal samples increase with the increase in the gas pressure. The larger the gas pressure, the larger the pore pressure and the greater the volumetric strain increase. In this process, gas adsorption is reduced to a secondary position.

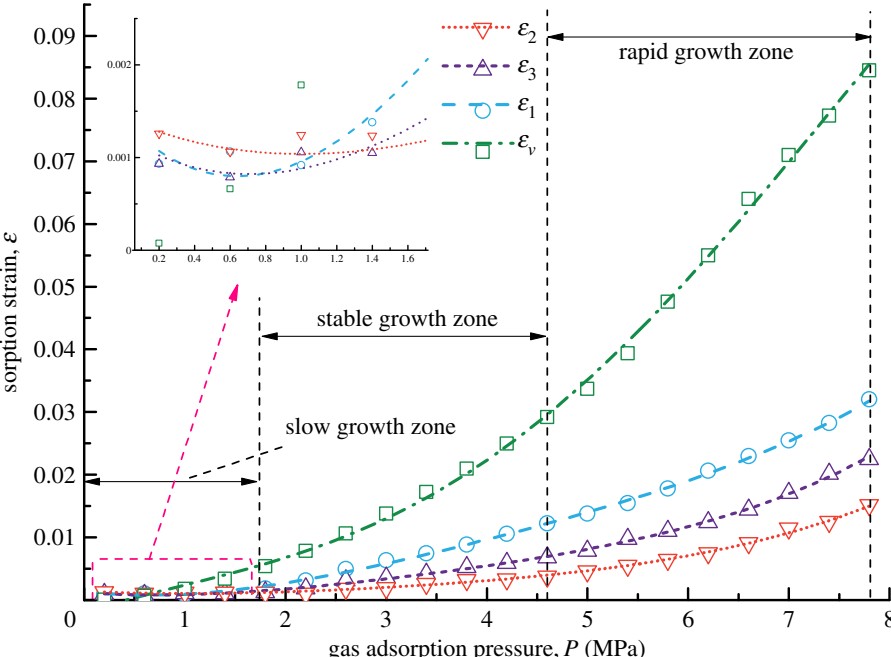

**Figure 7.** Deformation characteristics of Coal Sample 4 under different gas adsorption pressures.

**Table 2.** Increase in volumetric strains for coal samples.

| growth zone | sample | | | |
|---|---|---|---|---|
| | Coal Sample 1 | Coal Sample 2 | Coal Sample 3 | Coal Sample 4 |
| slow growth zone | 0.00084 | 0.00128 | 0.00079 | 0.00198 |
| stable growth zone | 0.01044 | 0.01420 | 0.01628 | 0.02721 |
| rapid growth zone | 0.02255 | 0.02652 | 0.02601 | 0.05529 |

## 3.2. Mechanism of gas adsorption deformation

In this paper, the density, Poisson's ratio, Young's modulus, bulk modulus, pore structure length, diameter and other parameters of soft coalbed methane reservoirs are adopted to model the coalbed methane reservoir medium [33]. In this model, the soft coalbed methane reservoir and gas satisfy the following conditions: (i) the reservoir is an isotropic continuous medium, which means that the physical and mechanical properties are the same in all directions; (ii) the reservoir is a saturated mixture composed of coal matrix skeleton and gas that is free and adsorbed in pore structures; (iii) the coal matrix is always in a solid state, and the adsorbed gas is always in a gaseous state, and the two will not transform into each other; (iv) the temperature of gas in reservoirs is kept constant during the adsorption and desorption process.

Gas in soft coalbed reservoirs mainly exists in an absorbed state, and the volume deformation of coal mass is caused by both pore pressure and gas adsorption. Schere [34] believed that when gas-containing coal is assumed to be an isotropic elastic medium, and the elasticity energy is equal to the change of surface energy, the expansion strain caused by gas adsorption is

$$\varepsilon_v = \gamma A \rho_s \frac{f(\varphi, v_s)}{E_s}, \tag{3.1}$$

$$f(\varphi, v_s) = \left(1 - \frac{4c\varphi(1 - 2v_s)}{3 - 5v_s}\right) \cdot \left(\frac{2(1 - v_s) - c\varphi(1 + v_s)}{2 - 3c\varphi}\right) \tag{3.1-1}$$

and

$$c = \frac{8\sqrt{2}}{3\pi}; \quad \varphi = \frac{a}{l}, \tag{3.1-2}$$

where $\varepsilon_v$ is the adsorption volumetric strain, $\gamma$ is the surface potential energy, $A$ is the specific surface area, $\rho_s$ is the density, $E_s$ is Young's modulus, $v_s$ is Poisson's ratio, $a$ is the pore radius and $l$ is the pore length.

Using the Langmuir adsorption model, Pan & Connell [35] obtained the relationship of surface potential energy with specific surface area and Langmuir adsorption constant with the help of the analysis of adsorption effects [36]

$$\gamma = \frac{\left[\int_0^p v^a \mathrm{d}p - RTP_L \ln(1 + V_L P)\right]}{A}.$$ (3.2)

With equations (3.1) and (3.2) combined, the relationship of the adsorption volumetric strain of coalbed methane with reservoir density, Young's modulus, Poisson's ratio, etc. is further simplified [37]

$$\varepsilon_v = \rho_s \frac{f(\varphi, v_s)}{E_s} \left[\int_0^p v^a \mathrm{d}p - RTP_L \ln(1 + V_L P)\right],$$ (3.3)

where $v_a$ is the volume of adsorbed gas, $P$ is the pressure of adsorbed gas, $R$ is the gas constant, $T$ is the temperature and $V_L$ and $P_L$ are the Langmuir constants.

Cui & Bustin [38] pointed out that the volumetric deformation of coal mass caused by gas adsorption is linear with the gas adsorption pressure

$$\varepsilon_v = \varepsilon_g \cdot v_a$$ (3.4)

and

$$v_a = \frac{V_L P}{P + P_L},$$ (3.4 − 1)

where $\varepsilon_g$ is the adsorption volumetric strain coefficient.

When the temperature of the physical experiment environment is constant, the influence of temperature on gas adsorption is not considered, and thus, the reservoir strain $\varepsilon_v$ caused by gas adsorption is

$$\varepsilon_v = \left(1 - \frac{4c\varphi(1 - 2v_s)}{3 - 5v_s}\right) \cdot \left(\frac{2(1 - v_s) - c\varphi(1 + v_s)}{2 - 3c\varphi}\right) \frac{\rho_s}{E_s} \int_{P_1}^{p_2} \frac{V_L \varepsilon_g P}{P + P_L} \mathrm{d}p.$$ (3.5)

After variable transformation and integration, the above equation is simplified

$$\varepsilon_v = \frac{V_L \rho_s \varepsilon_g f(\varphi, v_s)}{E_s} \cdot \left\{[P - P_L \ln(P + P_L)]\bigg|_{p_1}^{p_2}\right\}.$$ (3.6)

According to the gas adsorption deformation model described above, the adsorption deformation of a coalbed methane reservoir is affected by its own density, Poisson's ratio, Young's modulus, porosity, adsorbed gas pressure, etc. in a constant temperature physical experiment environment. After the corresponding values of the basic mechanical parameters of the coal samples in table 1 are put into equation (3.6), the theoretical curve of the gas adsorption deformation with the gas pressure is obtained, as shown in figures 8–11.

Figures 8–11 show the comparison of experimental and theoretical variation of the coal sample volumetric strain with the gas adsorption pressure. From the fitting curve of the modelled volumetric strain and the experimental volumetric strain for coal samples, it can be seen that the volumetric strain of coal samples increases gradually with the increase in the gas adsorption pressure. The fitting variances of the experimental and the theoretical results for the four kinds of coal samples are all above 98%, which indicates that the experimental results are very close to the theoretical results.

When the gas adsorption pressure is below Point A (2.5 MPa), the experimental volumetric strain of the coal samples is less than the theoretical volumetric strain. When the gas adsorption pressure is above Point B (about 5 MPa), the experimental volumetric strain is greater than the theoretical volumetric strain. When the gas adsorption pressure is between Point A and Point B (2–5 MPa), the experimental adsorption volumetric strain of the coal samples almost coincides with the theoretical adsorption volumetric strain curve.

The mechanism of influence of gas migration on the deformation of coal mass is analysed in combination with the gas adsorption deformation model (equation (3.6)). The gas in pores and fractures of coal mass is mainly in two states, an adsorbed state and a free state, and most of it is in an adsorbed state. The gas molecules are adsorbed on the pore surface and release adsorption heat energy causing coal mass to undergo expansion deformation. As the gas pressure increases, more gas molecules are adsorbed on the surface of coal, and more adsorption heat is released, finally resulting

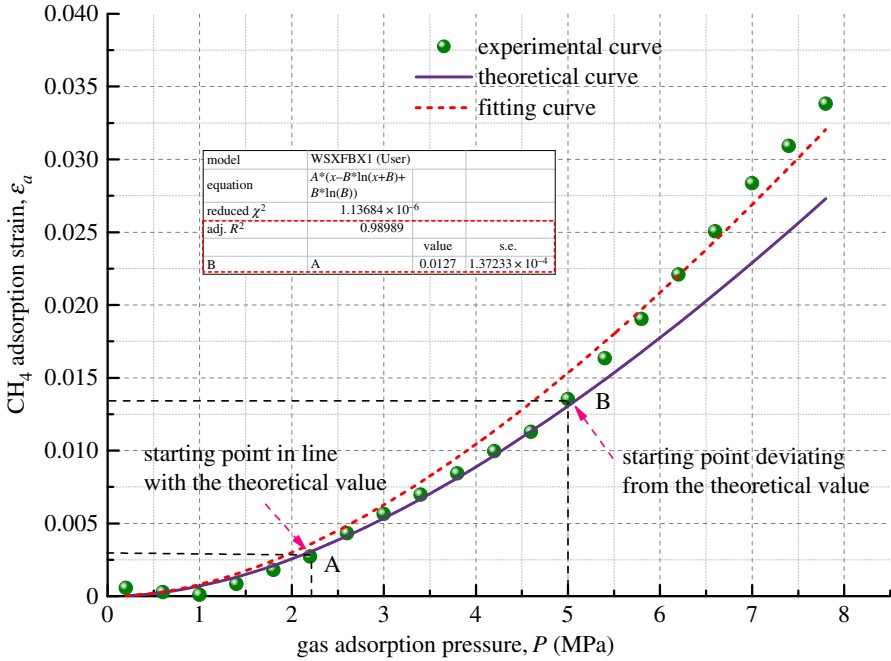

**Figure 8.** Relationship between volumetric strain and gas adsorption pressure for Coal Sample 1.

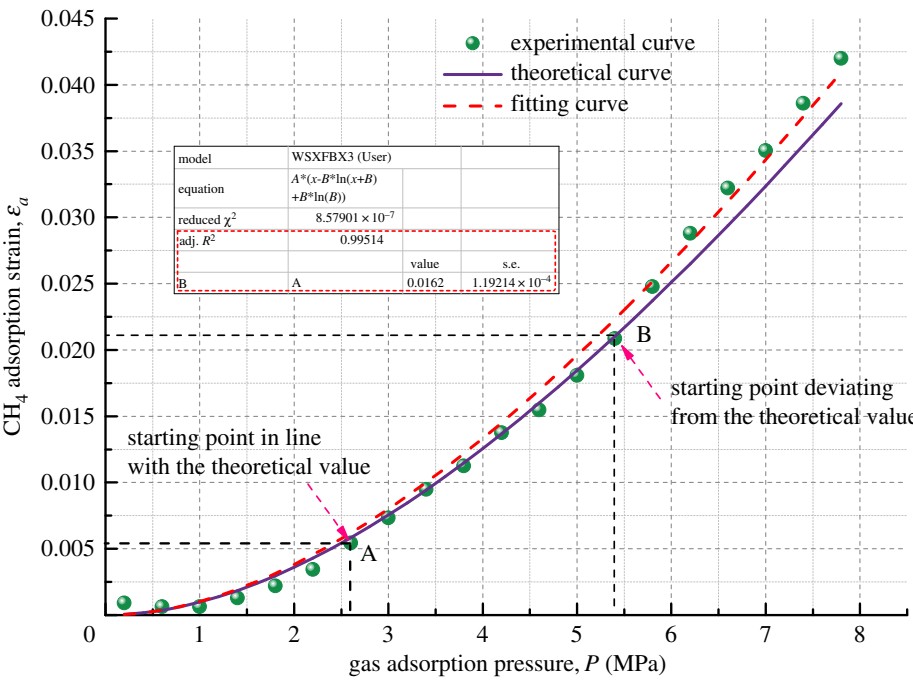

**Figure 9.** Relationship between volumetric strain and gas adsorption pressure for Coal Sample 2.

in larger expansion deformation. However, the amount of free gas in pores also increases substantially at the same time and forms an increasing pore pressure, which could result in deformation and failure of coal mass in the form of volumetric stress [39]. Therefore, when the gas pressure is large, the experimental volumetric strain results of coal mass are higher than the corresponding theoretical results. After the gas pressure is increased to a certain extent, the gas adsorption capacity is reduced when compared with that in the rapid gas adsorption deformation stage. This is because the gas adsorbed in the pore structure of the coal body is nearly saturated and cannot cause large adsorption deformation. Therefore, the adsorption deformation tends to be slow.

Furthermore, the fitting of the experimental results is performed. It is found that the fitting variance of the experimental and theoretical gas adsorption volumetric strain is above 98%, indicating that the

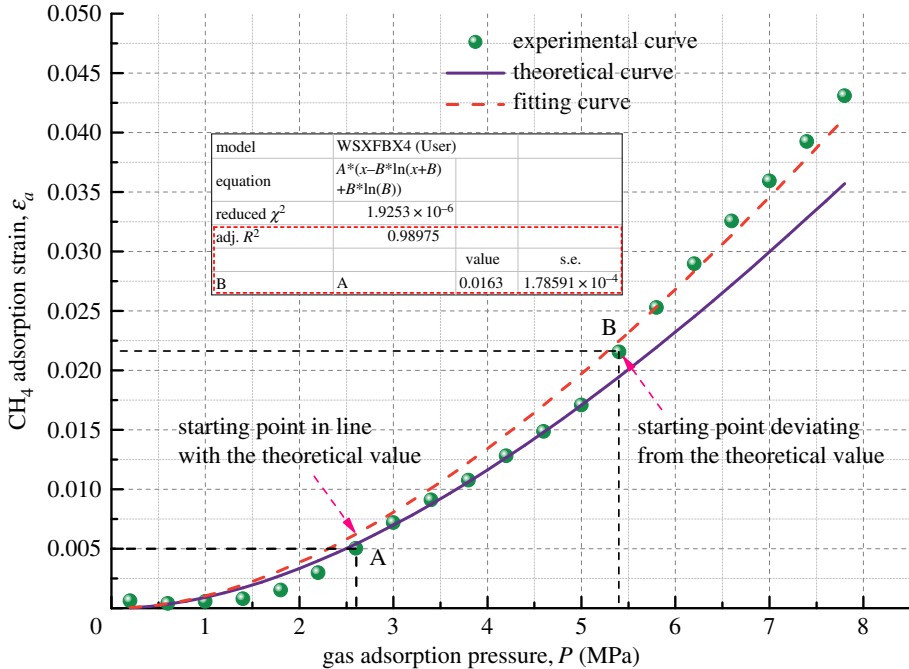

**Figure 10.** Relationship between volumetric strain and gas adsorption pressure for Coal Sample 3.

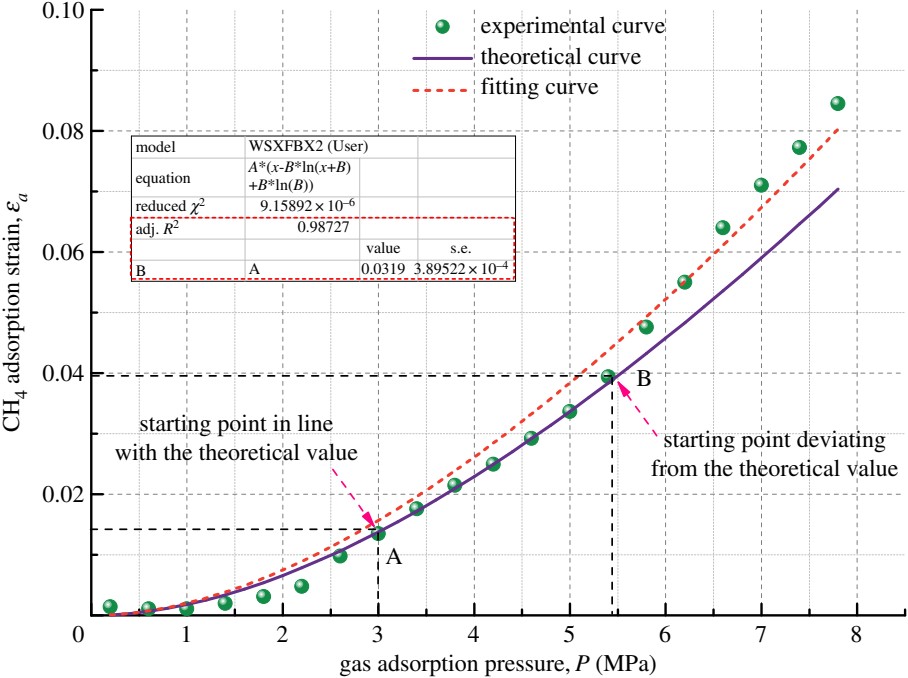

**Figure 11.** Relationship between volumetric strain and gas adsorption pressure for Coal Sample 4.

model developed in this paper can predict the volumetric strain variation of coal samples under different gas pressure conditions.

## 3.3. Influence of true triaxial stresses on permeability of soft coal mass

The permeability of the coal samples can be calculated from the gas pressure and flow rate measured in the experiment, and the equation for the calculation is [29]

$$K = \frac{2q\mu L P_n}{A(P_2^2 - P_1^2)}, \tag{3.7}$$

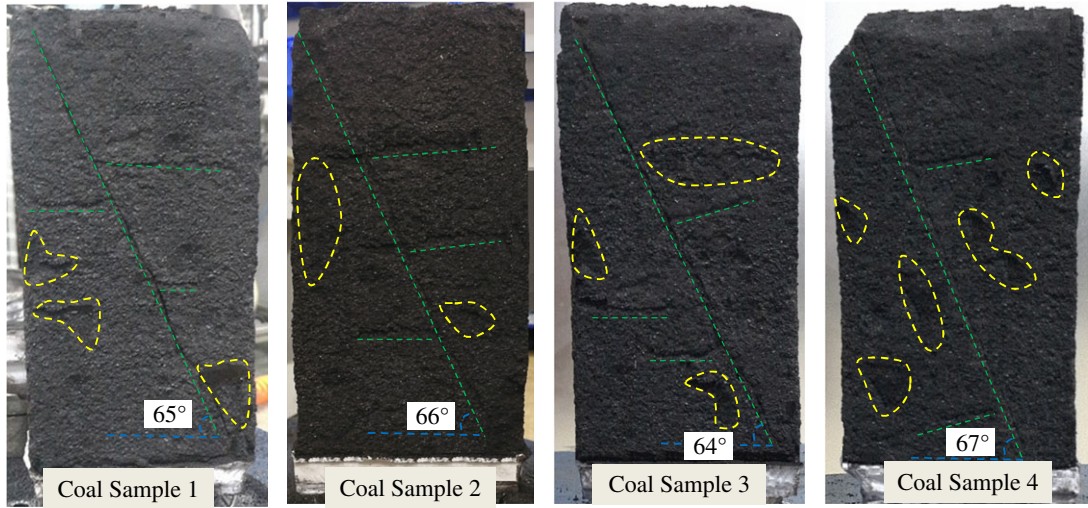

**Figure 12.** Coal sample forms after coal seepage experiments.

**Table 3.** Parameters of permeability experiments for coal samples.

| basic parameters | symbols and units (MPa) | Coal Sample 1 | Coal Sample 2 | Coal Sample 3 | Coal Sample 4 |
|---|---|---|---|---|---|
| CH$_4$ adsorption pressure | $P$ | 4 | 4 | 3 | 3 |
| minimum principal stress | $\sigma_3$ | 5 | 4.8 | 3.7 | 3.5 |
| intermediate principal stress | $\sigma_2$ | 7.5 | 7.2 | 5.6 | 5.3 |
| adsorption volumetric strain | $\varepsilon_v$ | 0.0089 | 0.0126 | 0.0070 | 0.0204 |

where $K$ is the permeability, m$^2$; $q$ is the gas percolation velocity of coal mass, m$^3$/s ($q = D \times C_k$, $D$ is the reading displayed on the flow accumulator, $C_k$ is the flowmeter coefficient, which is 0.719 for CH$_4$); $\mu$ is the gas dynamic viscosity coefficient ($1.10 \times 10^{-11}$ MPa s for CH$_4$); $L$ is the length of the sample, m; $P_n$ is 0.1 MPa; $A$ is the cross-sectional area, m$^2$; $P_1$ is the outlet gas pressure, MPa; $P_2$ is the inlet gas pressure, MPa.

In order to further study the relationship between coal permeability and stress, the differences in adsorption deformation and permeability of coal samples under different gas pressures are considered. According to the basic parameters of the four kinds of coal samples, the volumetric strains measured in the gas adsorption deformation experiment, and the theoretical model parameters of gas adsorption deformation, the parameters of the coal mass permeability experiment under true triaxial stress conditions are determined, as shown in table 3.

Figure 12 shows the forms of the four kinds of coal samples after the coal seepage experiment. In the coal seepage experiment under true triaxial stress conditions, the coal samples are mainly affected by shear stress, and obvious macroscopic fractures are formed on the surface of coal samples. The main fractures have a horizontal angle of about 65° (green dotted line), and flaky shedding pits are formed on the surface of the coal samples (yellow dotted line). Figure 13 shows the permeability variation curve of the four kinds of coal samples under true triaxial stress conditions. As the maximum principal stress difference ($\Delta\sigma$) increases under true triaxial stress conditions, the permeability of coal samples decreases gradually in different stages, and these stages are indicated by different zones: a slow reduction zone, a rapid reduction zone and a steady reduction zone.

For the slow reduction zone, with the increase of $\Delta\sigma$ from 2 MPa to about 4.5 MPa, the permeability of the four kinds of coal samples decreases to varying degrees. Specifically, it decreases by $0.79 \times 10^{-15}$ m$^2$ for Coal Sample 1, by $0.77 \times 10^{-15}$ m$^2$ for Coal Sample 2, by $0.43 \times 10^{-15}$ m$^2$ for Coal Sample 3 and by $0.71 \times 10^{-15}$ m$^2$ for Coal Sample 4, as shown in table 4. The permeability decreases slowly because the original pores of coal samples are continuously compressed, resulting in narrowed the seepage channels.

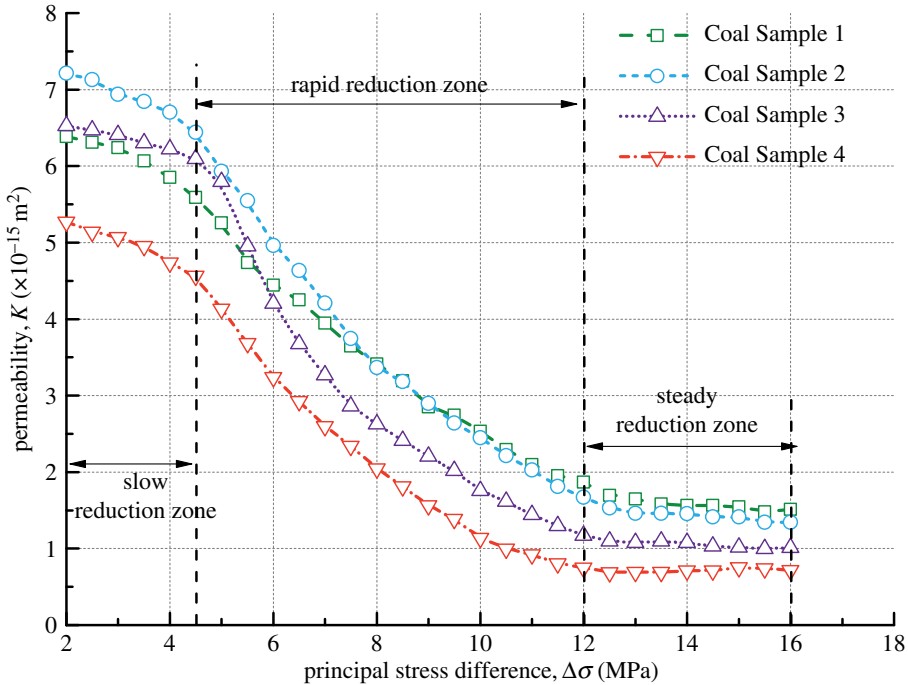

**Figure 13.** Variation of permeability with principal stress difference for four kinds of coal samples.

**Table 4.** Variation of permeability difference for coal samples.

| permeability ($\times 10^{-15}$ m$^2$) | Coal Sample 1 | Coal Sample 2 | Coal Sample 3 | Coal Sample 4 |
|---|---|---|---|---|
| initial permeability | 6.38 | 7.21 | 6.52 | 5.27 |
| slow reduction zone—permeability difference | 0.79 | 0.77 | 0.43 | 0.71 |
| rapid reduction zone—permeability difference | 3.72 | 4.77 | 4.92 | 3.81 |
| steady reduction zone—permeability difference | 0.36 | 0.33 | 0.16 | 0.03 |

For the rapid reduction zone, as $\Delta\sigma$ increases from 4.5 MPa to about 12 MPa, the permeability continues to decrease rapidly and approximately linearly, and the coal samples undergo elastic deformation. Now the coal samples enter the elastic stage and they are in a state of compression deformation. If the pressure relief measures are taken, the coal samples deformed can be restored to their original state. However, the original pores and fractures of coal samples continue to be compressed, and the permeability continues to decrease rapidly.

For the steady reduction zone, as $\Delta\sigma$ increases from about 12 MPa to 18 MPa, the coal samples are subjected to large true triaxial stresses. This leads to a transition of coal samples from a compressed state to an expanded state. New micro-fractures are generated owing to the relative slip between the internal particles of the coal samples, finally resulting in macro-fractures. However, the closure of the original fractures counteracts the generation of new fractures, causing the permeability of coal samples to decrease slowly. If $\sigma_1$ continues to be increased at this time, the coal samples are destructed after they reach their peak strength. As a consequence, the permeability of coal samples exhibits an upward trend.

## 3.4. Evolution mechanism of permeability for soft coal mass

Chikatamarla *et al.* [40] found through gas adsorption experiments that the volumetric strain of coal induced by gas adsorption is proportional to the amount of gas adsorbed. According to rock mechanics, the stress and strain of coal deformation can be expressed as [41]

$$\sigma_{ij} = \frac{E_s}{1+v_s}\left(\varepsilon_{ij} + \frac{v_s}{1-2v_s}\varepsilon_{cv}\delta_{ij}\right) + \xi p \delta_{ij} + K_s \varepsilon_v \delta_{ij} \qquad (3.8)$$

and

$$K_s = \frac{E_S}{3(1 - 2v_s)},\tag{3.8 - 1}$$

where $\varepsilon_{cv}$ is the volumetric strain of coal, $\varepsilon_v$ is the reservoir strain, $E_s$ is Young's modulus, $K_s$ is the bulk modulus, $\zeta$ is the biot coefficient with a range of 0–1, $\delta$ is the Kronecker delta function (when $i = j$, $\delta_{ij} = 1$; when $i \neq j$, $\delta_{ij} = 0$).

The area in front of the mining face of coalbed methane reservoirs is divided into a plastic zone, an elastoplastic zone, an elastic zone and a primary rock stress zone. Coal mass is under the three-dimensional stresses composed of vertical stress $\sigma_{zz}$ ($\sigma_1$), lateral stress $\sigma_{yy}$ ($\sigma_2$) and horizontal stress $\sigma_{xx}$ ($\sigma_3$). With $\varepsilon_3 = \varepsilon_2 = 0$ assumed, equation (3.8) combined, and the gas adsorption deformation model (equation (3.6)) considered, the numerical relationship between the initial three-dimensional stresses $\sigma_3$, $\sigma_2$ and $\sigma_1$ can be obtained

$$\sigma_3 = \frac{v_s}{1 - v_s}\sigma_1 + \frac{1 - 2v_s}{1 - v_s}\left[P + \frac{\rho_s f(\varphi, v_s)}{3(1 - v_s)}\int_{p_1}^{p_2}\frac{V_L \varepsilon_g P}{P + P_L}\,dp\right]\tag{3.9}$$

and

$$\varepsilon_1 = \frac{\sigma_1}{E_S};\quad \sigma_1 = \sigma_u;\quad \sigma_2 = (1 + \eta)\sigma_3,\tag{3.9 - 1}$$

where $\sigma_u$ is the uniaxial compressive strength of soft reservoir coal mass and $\eta$ is the length ratio of the standard sample of soft reservoir coal mass.

The change of the stress of reservoir coal in the vertical direction can be expressed as

$$\sigma_1 - \sigma_1^0 = \frac{2(1 - 2v_s)}{3(1 - v_s)}[(P - P_0) + K_s(\varepsilon_a - \varepsilon_{a0})].\tag{3.10}$$

According to Cui & Bustin [38] and Wang et al. [42], the relationship between stress, gas pressure and permeability is

$$k = k_0 \exp\left\{-\frac{3}{K_P}[(\sigma - \sigma_0) - (P - P_0)]\right\}.\tag{3.11}$$

Then, substitute equation (3.7) into equation (3.8)

$$k = k_0 \exp\left\{-\frac{3}{K_P}\left[\frac{(1 + v_s)\cdot(\sigma_1 - \sigma_1^0)}{2(1 - 2v_s)} - \frac{E_s(\varepsilon_a - \varepsilon_{a_0})}{3(1 - 2v_s)}\right]\right\},\tag{3.12}$$

where $K_P$ is the pore bulk modulus ($K_P = K_s \cdot \phi$), ($\sigma_1 - \sigma_1^0$) is the difference ($\Delta\sigma$) between $\sigma_1$ and the initial maximum principal stress ($\sigma_1^0$).

The gas adsorption volumetric strain difference ($\varepsilon_v$–$\varepsilon_{v0}$) can be solved by equation (2.6), and thus, the theoretical model for the permeability of coal mass changing with $\Delta\sigma$ under true triaxial stress conditions is obtained

$$k = k_0 \exp\left\{-\frac{3}{K_P}\left[\frac{(1 + v_s)\cdot(\sigma_1 - \sigma_1^0)}{2(1 - 2v_s)} - \frac{f(\varphi, v_s)V_L\rho_s\varepsilon_g}{3(1 - 2v_s)}\left(P - P_L\ln(P + P_L)\Big|_{p_1}^{p_2}\right)\right]\right\}.\tag{3.13}$$

The theoretical permeability variation curve of the coal samples can be obtained by equation (3.13) with the basic parameters of the coal samples in table 3 and the initial permeability measured in the seepage experiment. And then the theoretical curve and the experimental curve are compared, as shown in figures 14–17.

Figures 14–17 indicate that the experimental and theoretical permeability variations are consistent with the increase of $\Delta\sigma$ from 2 to 16 MPa in increments of 0.5 MPa when the gas permeation pressure difference is constant. They both show a trend of decreasing with increasing load stress, and the decrease in permeability gets slower and slower. The reason for this can be explained as follows. When the gas pressure is kept constant, the compressive stress that the coal samples are subjected to increases gradually with the increase in $\sigma_1$, leading to different degrees of the closure of the internal pores and fractures of coal samples. As a result, the porosity decreases, the fracture aperture decreases, the gas flow channel narrows and the permeability of coal samples decreases.

Based on the parameters of density, Poisson's ratio, Young's modulus, bulk modulus and initial permeability of different soft coal samples, the permeability model developed in this paper can

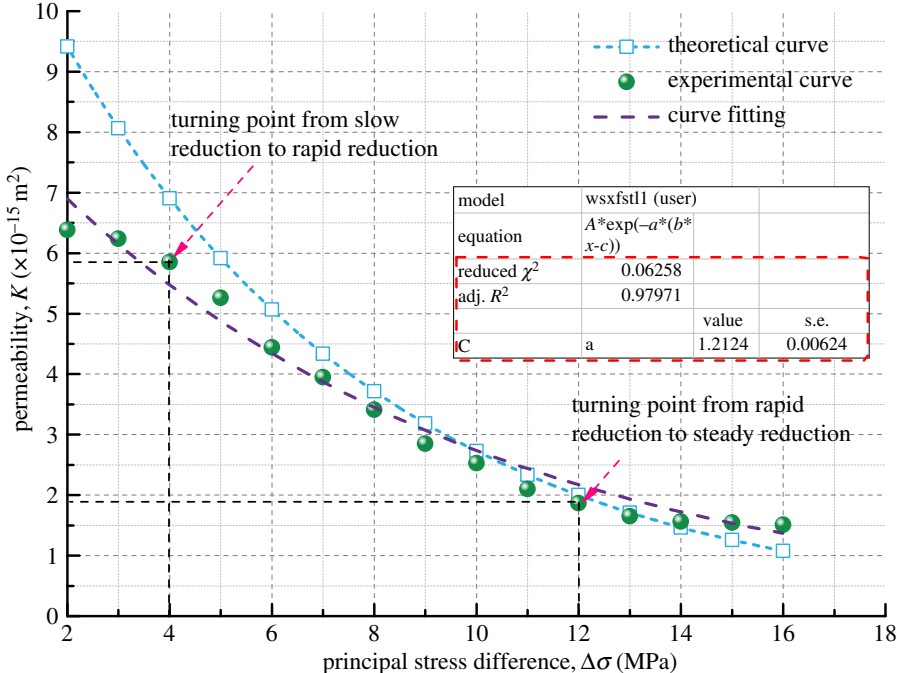

**Figure 14.** Variation of permeability with principal stress difference for Coal Sample 1.

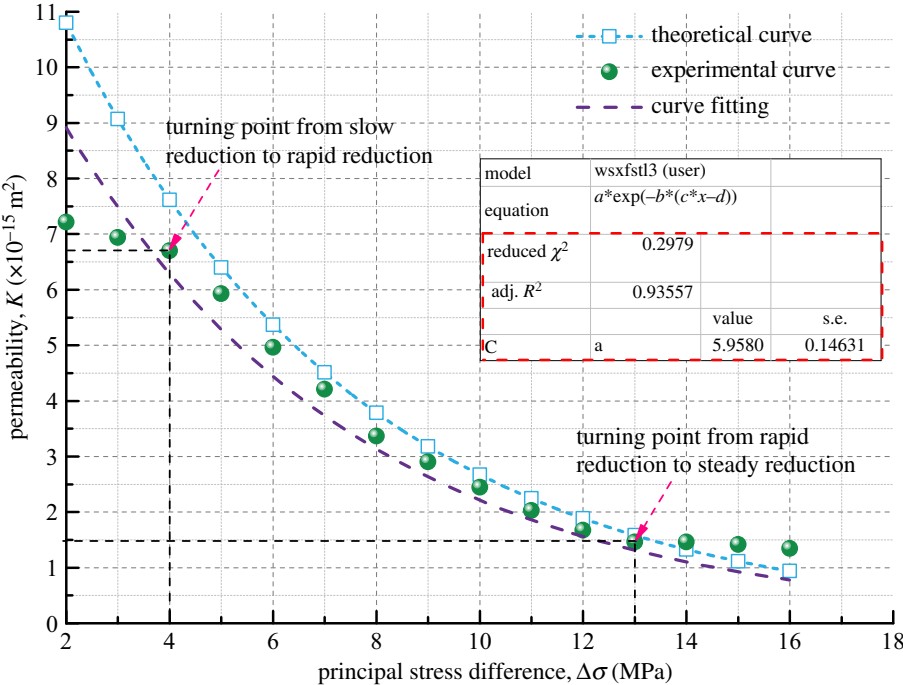

**Figure 15.** Variation of permeability with principal stress difference for Coal Sample 2.

predict the permeability of coal in the rapid reduction zone pretty well, but it is not enough to reflect the characteristics of the transition between the three zones. However, the overall fitting variance of the theoretical model results and the experimental results is between 91 and 97%, indicating that the coal permeability variation model has a high accuracy and a good applicability.

## 4. Discussion

The gas adsorption experiment and permeability experiment of coal in soft coalbed methane reservoirs under true triaxial stress conditions are carried out, and the gas adsorption deformation model and coal

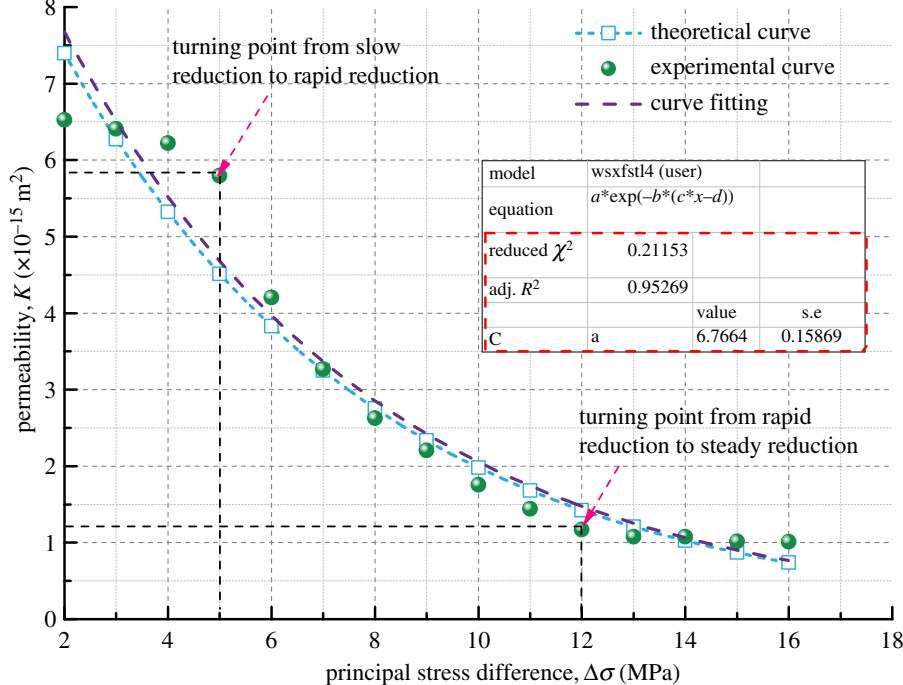

**Figure 16.** Variation of permeability with principal stress difference for Coal Sample 3.

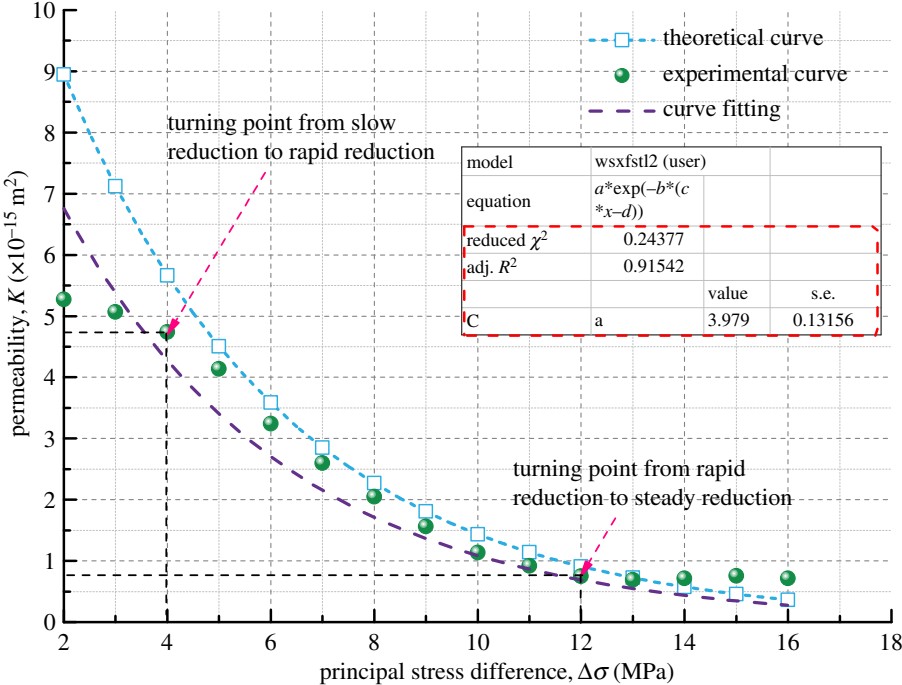

**Figure 17.** Variation of permeability with principal stress difference for Coal Sample 4.

permeability model are proposed in this paper. These models take the mechanism of gas adsorption and migration of soft coalbed methane reservoirs into account. However, the influence of single factors such as density, Poisson's ratio and uniaxial compressive strength on the gas adsorption and seepage of soft coalbed methane reservoirs is not quantitatively analysed. Only the overall deformation and permeability trends of different soft coal samples are examined. After the analysis of the adsorption deformation of the four kinds of coal samples, it is indicated in figures 4–7 that the strains of Coal Sample 4 are smaller than those of the other three coal samples while the Poisson ratio and porosity of Coal Sample 4 are larger than those of the other three coal samples. In addition, the other parameters such as density, Young's modulus,

and bulk modulus of all four coal samples are similar. Therefore, the reason for the small adsorption deformation of the coal samples might be the large Poisson's ratio and porosity of the coal samples. In the future, the orthogonal adsorption-seepage experiments on the different mechanical parameters of the coal samples involved in this paper will be carried out systematically in order to further analyse the applicable range of the coal permeability model with gas adsorption taken into account.

## 5. Conclusion

In this paper, the influence of gas adsorption and migration on the deformation and permeability of soft coal mass under true triaxial stress conditions is explored through the self-developed true triaxial 'gas–solid' coupled coal mass seepage experimental system. Furthermore, the gas adsorption deformation model and permeability model of coal mass are developed. In addition, the mechanism behind the gas adsorption-seepage experimental phenomenon is investigated. From all the analysis, the following main conclusions are drawn:

(1) In the coal mass adsorption and seepage experiment under true triaxial stress conditions, the gas adsorption deformation variation of soft coal mass with the increase in gas pressure is divided into a slow growth zone, a stable growth zone and a rapid growth zone. The permeability variation is divided into a slow reduction zone, a rapid reduction zone and a steady reduction zone with two obvious turning points as the principal stress difference increases.

(2) The deformation characteristics of the soft reservoir coal during the gas adsorption experiment under true triaxial stress conditions are analysed theoretically and the gas adsorption deformation model is developed. This model can predict the dynamic evolution trend of the adsorption deformation of soft coal samples under different gas adsorption pressures because the fitting variance of the theoretical and experimental results is above 98%.

(3) Through the gas adsorption deformation model and the permeability experiment of the coal samples under true triaxial stress conditions, the relationship between the gas adsorption deformation and permeability of the soft coal samples is further analysed, and the permeability evolution model of coal mass considering the gas adsorption is developed. It is found that the fitting variance of the theoretical and experimental permeabilities is above 91%, and that this model can predict the variation trend of coal mass permeability in the rapid reduction zone.

Data accessibility. The datasets supporting this article have been uploaded as part of the electronic supplementary material.

Authors' contributions. W.W. and J.L. conceived and designed the experiments and theoretical models; Y.H. and C.F. performed the experiments; G.W. and Z.L. interpreted the results and wrote the manuscript. All authors gave final approval for publication.

Competing interests. We declare we have no competing interests.

Funding. This research was funded by the National Natural Science Foundation of China (grant no. 51674158), Shandong University of Science and Technology Outstanding Youth Science and Technology Talent Support Program (grant no. 2015JQJH105), Qingdao Applied Basic Research Project (no. 17-1-1-38-jch).

Acknowledgement. Jian Chen and Yue Wang helped with experimental work and their efforts are highly appreciated. Hao Xu is acknowledged for his contribution to the preparation of thin sections. Xuelin Liu's technical support is also greatly appreciated.

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
