## [Reviewer comments · Royal Society Open Science]

Review History

RSOS-190892.R0 (Original submission)

Review form: Reviewer 1

Is the manuscript scientifically sound in its present form?

No

Are the interpretations and conclusions justified by the results?

No

Is the language acceptable?

No

Do you have any ethical concerns with this paper?

No

Have you any concerns about statistical analyses in this paper?

Yes

Recommendation?

Major revision is needed (please make suggestions in comments)

Comments to the Author(s)

A number of experimental measurements have been undertaken on Coal samples looking at the impact of gas sorption and changes in permeability related to strain. The changes in strain and the permeability changes are then modelled using a combination of rock mechanics deformation models, a sorption model and a permeability model. The paper is well presented.

As there is only a "yes // no" option to a number of the questions I have been asked, I have been forced to choose the "no" option, where in most cases it is close to being acceptable. However there are a number of key issues which need to be clarified or revisited. I have attached a commented manuscript which will aid the authors in reviewing their paper.

A key issue is that the authors identify a "slow", "stable" and "Rapid" sample growth zone, based on the relationship to an elastic model describing the relationship between gas pressure and sample strain. However there is no clear inclusion of the triaxial state of pressure in this model, and it appears that a formulation is used that uses a value of stress as the surface potential energy. The issue is that the division of stable to rapid is occurring when the gas pressure is significantly above two of the axial pressures. The model is not able to account for the triaxial stress, and thus I believe this is significantly influencing the results and the interpretation. i.e. the model used is not really valid for this type of interpretation. I would suggest revisiting the results with this in mind, as the experimental results are certainly good quality. Also the modelling results show good correlations..however process understanding is key.

Likewise the permeability model appears to be assuming no axial strain in the s_2 or s_3 direction, however the experimental results show strain in these directions.

I have made a number of more detailed comments on the way through the manuscript.

The permeability model developed appears principally to be a 1D s_1 approximation, which may explain the slightly less accurate fitting of the experimental results.

The results where gas pressure above a principal axis stress has been allowed need to be revisited, as there is then no real control on strain.

Although the experimental work is truly triaxial, it appears the modelling is not. This is fine as long as the limitations and best approximations are included and an understanding of what the limiting factors may be.

The title needs rethinking a bit as it doesn't really reflect the contents, currently emphasises the flow aspect of gas, but the work is all about gas pressures and stress conditions.

Review form: Reviewer 2

Is the manuscript scientifically sound in its present form?

Yes

Are the interpretations and conclusions justified by the results?

Yes

Is the language acceptable?

Yes

Do you have any ethical concerns with this paper?

No

Have you any concerns about statistical analyses in this paper?

No

Recommendation?

Accept with minor revision (please list in comments)

Comments to the Author(s)

The author designed a gas adsorption and seepage test for soft coal under true triaxial stress conditions for soft coal. The stress conditions are in line with coal mining. The true three-dimensional stress state in the process, the research object and the research method are very innovative. Based on the true triaxial test, the author established a theoretical model of coal gas adsorption deformation and a model of seepage evolution. The results of the two models are similar to those exhibited by the true triaxial test. It is the verification of the accuracy of the true triaxial test and the verification of the applicability of the adsorption and seepage models. The analysis is clear and logical. Therefore, the research results of this paper are very valuable, and can provide experimental and theoretical guidance for coalbed methane extraction, and it is recommended to accept after minor revision.

1. The true triaxial seepage test system of this paper is independently developed by the author. I am interested in this. I propose a design pattern of "two rigid and one soft, three-way independent loading". Why should we design this way? What are the advantages? Can the author explain?
2. The manuscript text on page 2, line 34 is not used properly. The author says "the permeation law of shale". This is not accurate. It is recommended that the author change to "Seepage Regulation of shale".
3. The adsorption and percolation tests in this paper all require high sealing performance of the test equipment, and the author does not explain how the coal sample is sealed. Is it because of the length of the article? If yes, please explain here; if not, please add the reason to the body.
4. The author of Figure 1 is very beautiful, but there are three other signs in the lower left corner. If it is related to the device, please mark it. If not, please delete it.
5. From the author's industrial analysis of coal samples, it is known that raw coal is relatively soft and not easy to form. The author solved this problem by using coal briquettes. This method is feasible and worth promoting. But does the author consider the consistency of other mechanical properties of briquette with raw coal?
6. In Figure 3, the first three coal samples are the same color, and the fourth coal sample has a high exposure. Can the author be replaced by a uniform one? Is there no uniform exposure after the end of the test?
7. It can be seen from Fig. 4-7 that when the gas adsorption pressure is lower than 1 MPa, the deformation of the coal sample in three directions is inconsistent, and the variation law is not uniform. According to the experience of reviewing experts, this stage is indeed complicated. The deformation of coal samples is dominated by stress or gas adsorption, and even the creep characteristics of coal samples. There is no unified understanding. How does the author understand the mechanism of action of gas adsorption at lower pressures?
8. In the manuscript, on page 10, line 22, the author points out that "the coal mass after its gas adsorption is more likely to be deformed than before its gas adsorption". This view is correct, but for the language to be more concise, it is recommended that the author "before" Delete later.

9. Similarly, on page 10, line 52, the author wants to express that the higher the adsorption pressure, the higher the amount of gas adsorbed? The authors say "and more gas is adsorbed on the pore surface of the coal samples resulting in a larger adsorption thickness." The suggestion is changed to "and more gas is adsorbed on the pore surface of the coal samples, resulting in a larger adsorption thickness."
10. On page 11, page 28 of the manuscript, the three stress directions are mistake, and there are two periods at the end of the paragraph. I hope the author can check all the writing formats in the text.
11. What does "model:WSXFBX1(USER)" in Figure 8 mean? The same as Figure 9-11 below, is the fitted model? Is it different from the theoretical model derived by the author?
12. On page 15, line 15 of the manuscript, the unit of gas flow is error.
13. There is a problem with the format of Table 3.
14. The format of the headings of Figures 10 and 11 is not uniform.
15. From the true triaxial adsorption test in Figure 8-11, two turning points can be clearly observed, but the theoretical model curve does not show a turning point. The reason is also explained in detail in the article, but the author can correct the model parameters and make the theory Is the model better matched to the test results?
16. The unit of density in Table 4 is not written correctly.
17. Is the true triaxial seepage test in this paper based on the previous adsorption test? Or is the two tests separate? From the author's point of view, the two tests are carried out separately, so that the coal sample failure morphology after the true triaxial gas adsorption can be obtained, and the author is established to conduct the continuity test. If it is a continuous test, is the form of coal sample adsorption deformation obtained in a separate study of gas adsorption?
18. Like the true triaxial adsorption test, the seepage test also showed a turning point, which is similar to the theoretical model of seepage. Can the author also make parameter corrections to keep the two closer?

Decision letter (RSOS-190892.R0)

07-Aug-2019

Dear Dr Wang,

The editors assigned to your paper ("Influence of Gas Migration on Permeability of Soft Coalbed Methane Reservoirs under True Triaxial Stress Conditions") have now received comments from reviewers. We would like you to revise your paper in accordance with the referee and Associate Editor suggestions which can be found below (not including confidential reports to the Editor). Please note this decision does not guarantee eventual acceptance.

Please submit a copy of your revised paper before 30-Aug-2019. Please note that the revision deadline will expire at 00.00am on this date. If we do not hear from you within this time then it will be assumed that the paper has been withdrawn. In exceptional circumstances, extensions may be possible if agreed with the Editorial Office in advance. We do not allow multiple rounds of revision so we urge you to make every effort to fully address all of the comments at this stage. If deemed necessary by the Editors, your manuscript will be sent back to one or more of the original reviewers for assessment. If the original reviewers are not available, we may invite new reviewers.

To revise your manuscript, log into <http://mc.manuscriptcentral.com/rsos> and enter your Author Centre, where you will find your manuscript title listed under "Manuscripts with

Decisions." Under "Actions," click on "Create a Revision." Your manuscript number has been appended to denote a revision. Revise your manuscript and upload a new version through your Author Centre.

- Data accessibility

If you wish to submit your supporting data or code to Dryad (<http://datadryad.org/>), or modify your current submission to dryad, please use the following link:
<http://datadryad.org/submit?journalID=RSOS&manu=RSOS-190892>

- Competing interests

- Authors' contributions

AB carried out the molecular lab work, participated in data analysis, carried out sequence alignments, participated in the design of the study and drafted the manuscript; CD carried out the statistical analyses; EF collected field data; GH conceived of the study, designed the study,

coordinated the study and helped draft the manuscript. All authors gave final approval for publication.

- Acknowledgements

- Funding statement

Kind regards,

on behalf of Professor Rachel Wood (Associate Editor) and R. Kerry Rowe (Subject Editor)
openscience@royalsociety.org

Comments to Author:

Reviewers' Comments to Author:

Reviewer: 1

Comments to the Author(s)

A number of experimental measurements have been undertaken on Coal samples looking at the impact of gas sorption and changes in permeability related to strain. The changes in strain and the permeability changes are then modelled using a combination of rock mechanics deformation models, a sorption model and a permeability model. The paper is well presented.

As there is only a "yes // no" option to a number of the questions I have been asked, I have been forced to choose the "no" option, where in most cases it is close to being acceptable. However there are a number of key issues which need to be clarified or revisited. I have attached a commented manuscript which will aid the authors in reviewing their paper.

A key issue is that the authors identify a "slow", "stable" and "Rapid" sample growth zone, based on the relationship to an elastic model describing the relationship between gas pressure and sample strain. However there is no clear inclusion of the triaxial state of pressure in this model, and it appears that a formulation is used that uses a value of stress as the surface potential energy. The issue is that the division of stable to rapid is occurring when the gas pressure is significantly above two of the axial pressures. The model is not able to account for the triaxial stress, and thus I believe this is significantly influencing the results and the interpretation. i.e. the model used is not really valid for this type of interpretation. I would suggest revisiting the results with this in mind, as the experimental results are certainly good quality. Also the modelling results show good correlations..however process understanding is key.

Likewise the permeability model appears to be assuming no axial strain in the s_2 or s_3 direction, however the experimental results show strain in these directions.

I have made a number of more detailed comments on the way through the manuscript.

The permeability model developed appears principally to be a 1D s_1 approximation, which may explain the slightly less accurate fitting of the experimental results.

The results where gas pressure above a principal axis stress has been allowed need to be revisited, as there is then no real control on strain.

Although the experimental work is truly triaxial, it appears the modelling is not. This is fine as long as the limitations and best approximations are included and an understanding of what the limiting factors may be.

The title needs rethinking a bit as it doesn't really reflect the contents, currently emphasises the flow aspect of gas, but the work is all about gas pressures and stress conditions.

Reviewer: 2

Comments to the Author(s)

The author designed a gas adsorption and seepage test for soft coal under true triaxial stress conditions for soft coal. The stress conditions are in line with coal mining. The true three-dimensional stress state in the process, the research object and the research method are very innovative. Based on the true triaxial test, the author established a theoretical model of coal gas adsorption deformation and a model of seepage evolution. The results of the two models are similar to those exhibited by the true triaxial test. It is the verification of the accuracy of the true triaxial test and the verification of the applicability of the adsorption and seepage models. The analysis is clear and logical. Therefore, the research results of this paper are very valuable, and can provide experimental and theoretical guidance for coalbed methane extraction, and it is recommended to accept after minor revision.

1. The true triaxial seepage test system of this paper is independently developed by the author. I am interested in this. I propose a design pattern of "two rigid and one soft, three-way independent loading". Why should we design this way? What are the advantages? Can the author explain?
2. The manuscript text on page 2, line 34 is not used properly. The author says "the permeation law of shale". This is not accurate. It is recommended that the author change to "Seepage Regulation of shale".
3. The adsorption and percolation tests in this paper all require high sealing performance of the test equipment, and the author does not explain how the coal sample is sealed. Is it because of the length of the article? If yes, please explain here; if not, please add the reason to the body.
4. The author of Figure 1 is very beautiful, but there are three other signs in the lower left corner. If it is related to the device, please mark it. If not, please delete it.
5. From the author's industrial analysis of coal samples, it is known that raw coal is relatively soft and not easy to form. The author solved this problem by using coal briquettes. This method is

feasible and worth promoting. But does the author consider the consistency of other mechanical properties of briquette with raw coal?

6. In Figure 3, the first three coal samples are the same color, and the fourth coal sample has a high exposure. Can the author be replaced by a uniform one? Is there no uniform exposure after the end of the test?

7. It can be seen from Fig. 4-7 that when the gas adsorption pressure is lower than 1 MPa, the deformation of the coal sample in three directions is inconsistent, and the variation law is not uniform. According to the experience of reviewing experts, this stage is indeed complicated. The deformation of coal samples is dominated by stress or gas adsorption, and even the creep characteristics of coal samples. There is no unified understanding. How does the author understand the mechanism of action of gas adsorption at lower pressures?

8. In the manuscript, on page 10, line 22, the author points out that "the coal mass after its gas adsorption is more likely to be deformed than before its gas adsorption". This view is correct, but for the language to be more concise, it is recommended that the author "before" Delete later.

9. Similarly, on page 10, line 52, the author wants to express that the higher the adsorption pressure, the higher the amount of gas adsorbed? The authors say "and more gas is adsorbed on the pore surface of the coal samples resulting in a larger adsorption thickness." The suggestion is changed to "and more gas is adsorbed on the pore surface of the coal samples, resulting in a larger adsorption thickness."

10. On page 11, page 28 of the manuscript, the three stress directions are mistake, and there are two periods at the end of the paragraph. I hope the author can check all the writing formats in the text.

11. What does "model:WSXFBX1(USER)" in Figure 8 mean? The same as Figure 9-11 below, is the fitted model? Is it different from the theoretical model derived by the author?

12. On page 15, line 15 of the manuscript, the unit of gas flow is error.

13. There is a problem with the format of Table 3.

14. The format of the headings of Figures 10 and 11 is not uniform.

15. From the true triaxial adsorption test in Figure 8-11, two turning points can be clearly observed, but the theoretical model curve does not show a turning point. The reason is also explained in detail in the article, but the author can correct the model parameters and make the theory Is the model better matched to the test results?

16. The unit of density in Table 4 is not written correctly.

17. Is the true triaxial seepage test in this paper based on the previous adsorption test? Or is the two tests separate? From the author's point of view, the two tests are carried out separately, so that the coal sample failure morphology after the true triaxial gas adsorption can be obtained, and the author is established to conduct the continuity test. If it is a continuous test, is the form of coal sample adsorption deformation obtained in a separate study of gas adsorption?

18. Like the true triaxial adsorption test, the seepage test also showed a turning point, which is similar to the theoretical model of seepage. Can the author also make parameter corrections to keep the two closer?

Author's Response to Decision Letter for (RSOS-190892.R0)

See Appendices A & B.

Decision letter (RSOS-190892.R1)

10-Sep-2019

Dear Dr Wang,

I am pleased to inform you that your manuscript entitled "Influence of Gas Migration on Permeability of Soft Coalbed Methane Reservoirs under True Triaxial Stress Conditions" is now accepted for publication in Royal Society Open Science.

on behalf of Professor Rachel Wood (Associate Editor) and R. Kerry Rowe (Subject Editor)
openscience@royalsociety.org

Appendix A**ROYAL SOCIETY
OPEN SCIENCE****Influence of Gas Migration on Permeability of Soft Coalbed
Methane Reservoirs under True Triaxial Stress Conditions**

Journal:	Royal Society Open Science
Manuscript ID	RSOS-190892
Article Type:	Research
Date Submitted by the Author:	01-Jun-2019
Complete List of Authors:	Wang, Gang; Shandong University of Science and Technology, College of Mining and Safety Engineering liu, zhiyuan; Shandong University of Science and Technology, College of Mining and Safety Engineering hu, yanwei; Shandong University of Science and Technology, College of Mining and Safety Engineering fan, cheng; Shandong University of Science and Technology, College of Mining and Safety Engineering wang, wenrui; Shandong University of Science and Technology, College of Mining and Safety Engineering li, jinzhou; Shandong University of Science and Technology, College of Mining and Safety Engineering
Subject:	Energy < ENGINEERING AND TECHNOLOGY, Engineering geology < ENGINEERING AND TECHNOLOGY, Environmental engineering < ENGINEERING AND TECHNOLOGY
Keywords:	True triaxial stress, Soft coalbed, Gas migration, Permeability model
Subject Category:	Engineering

Author-supplied statements

Relevant information will appear here if provided.

Ethics

Does your article include research that required ethical approval or permits?:

This article does not present research with ethical considerations

Statement (if applicable):

CUST_IF_YES_ETHICS :No data available.

Data

It is a condition of publication that data, code and materials supporting your paper are made publicly available. Does your paper present new data?:

Yes

Statement (if applicable):

The datasets supporting this article have been uploaded as part of the electronic supplementary material.

Conflict of interest

I/We declare we have no competing interests

Statement (if applicable):

CUST_STATE_CONFLICT :No data available.

Authors' contributions

This paper has multiple authors and our individual contributions were as below

Statement (if applicable):

W.R.W. and J.Z.L. conceived and designed the experiments and theoretical models; Y.W.H. and F.C. performed the experiments; G.W. and Z.Y.L. interpreted the results and wrote the manuscript. All authors gave final approval for publication.

Influence of Gas Migration on Permeability of Soft Coalbed Methane Reservoirs under True Triaxial Stress Conditions

Gang Wang^{a, b, *}, Zhiyuan Liu^b, Yanwei Hu^b, Cheng Fan^b, Wenrui Wang^b, Jinzhou Li^b

^a Shandong University of Science and Technology, Mine Disaster Prevention and Control-Ministry of State Key Laboratory Breeding Base, Qingdao 266590, China

^b Shandong University of Science and Technology, College of Mining and Safety Engineering, Qingdao 266590, China

Abstract: The permeability of coal body is the key parameter restricting the efficient extraction of coalbed methane, and scholars have analyzed it from two angles of the change of stress state and porosity of coal body. However, there is still a lack of study on the mechanism of gas migration and movement in soft coalbed methane reservoir under the coupling between the true triaxial stress field (maximum principal stress $\sigma_1 >$ intermediate principal stress $\sigma_2 >$ minimum principal stress σ_3) and the gas pressure field. In this paper, the coal gas adsorption and seepage experiments are conducted through the self-developed true triaxial "gas-solid" coupled coal mass seepage system with gas as the adsorption and seepage medium and coal briquette taking the place of soft coalbed methane reservoirs. Furthermore, the coal gas adsorption deformation model and the permeability evolution model taking gas adsorption into account are developed. Through analysis of both experimental and theoretic results, main conclusions are drawn as follows: (1) With the increase of gas pressure, the adsorption deformation variation of coal mass is divided into a slow growth zone, a stable growth zone and a rapid growth zone. (2) The gas adsorption deformation model developed can predict the variation trend of coal mass adsorption volumetric strains for different types of soft coalbeds, and the fitting variance of experimental and theoretical volumetric strains is above 98%. (3) With the increase of maximum principal stress difference, the coal permeability variation curve shows 2 obvious turning points, which can be divided into a slow reduction zone, a rapid reduction zone and a steady reduction zone. (4) The permeability model of coal mass considering the gas adsorption effect can reflect the variation characteristics of permeability in the rapid reduction zone, and the overall fitting variance of experimental and theoretical permeabilities is above 91%. The above results could provide a reliable experimental and theoretical basis for improving coalbed methane extraction rates.

Key words: True triaxial stress, Soft coalbed, Gas migration, Permeability model

1 Introduction

During coal seam mining process, coal mass in front of working faces is usually affected by mining disturbances and thus subject to unequal stresses in three directions. To be specific, the support pressure in the vertical direction increases and the pressure in the horizontal directions is relieved. As a consequence, expansion deformation of coal mass ars (Wang and Pang, 2017), and further continuous development, expansion and penetration of pores and fracture structures in coal mass occur, finally leading to desorption, permeability enhancement and migration of gas in pores (Xie et al., 2013). Especially in the presence of soft coalbeds, pore fracture development and gas migration become more severe, which have a dynamic influence on the stable extraction of coalbed methane. Therefore, understanding the mechanism of deformation damage and permeability evolution of soft coal mass under the coupling of gas and stress is essential to increase coalbed methane extraction rates (Pan and Connell, 2012) and realize scientific coal mining (Qian et al., 2018).

Many researchers have studied the adsorption of coalbed methane and the permeability of coal and rock mass through physical experiments. Meng and Li, (2017) and Connell et al. (2016) studied the effect of gas adsorption on coal matrix and cleat deformation. In addition, some researchers focused on the permeability characteristics of coal mass and conducted a large number of physical experiments (Geng et al., 2017; Pan et al., 2018; Liu et al., 2018; Mitra et al., 2012; Yin et al., 2015). A few researchers investigated the permeation law 
[revised manuscript text omitted]

(1) Gas adsorption path for coal samples (2) Permeability evolution path for coal samples

Figure 2 Schematic diagram of experimental method and procedure

3 Test results and analysis

3.1 Effect of gas adsorption on deformation of soft coal mass

Figure 3 shows the forms of 4 kinds of coal samples after the gas adsorption experiments. From the appearance of the coal samples, no obvious difference in their deformation forms are found. Figure 4-7 shows the deformation curves of the coal samples caused by gas adsorption and measured under true triaxial stress conditions. When the external stress is kept constant, the gas adsorption pressure increases from 0 to 8 MPa in increments of 0.2 MPa. As the gas adsorption pressure increases, the strains ($\epsilon_1, \epsilon_2, \epsilon_3$) of the 4 kinds of coal samples in the σ_1, σ_2 and σ_3 directions increase gradually, and the

volumetric strain (ϵ_v) also increases gradually. However, for each kind of coal sample, ϵ_v is the maximum, followed by ϵ_1 and then ϵ_3 , and ϵ_2 is the minimum, i.e., $\epsilon_v > \epsilon_1 > \epsilon_3 > \epsilon_2$.

Figure 3 Forms of coal samples after gas adsorption experiments

Figure 4 Deformation characteristics of Coal Sample 1 under different gas adsorption pressures

Figure 5 Deformation characteristics of Coal Sample 2 under different gas adsorption pressures

Figure 6 Deformation characteristics of Coal Sample 3 under different gas adsorption pressures

Figure 7 Deformation characteristics of Coal Sample 4 under different gas adsorption pressures

From Figure 4-7, it can be seen that the deformation variation of coal mass exhibits different deformation rates under different gas adsorption pressures, and that it can be divided into a slow growth zone, a stable growth zone and a rapid growth zone. In the slow growth zone, when the gas adsorption pressure is lower than about 1 MPa, the strains in all three directions and volumetric strains of the coal samples change slightly. For example, the volumetric strain increases by 0.00084 for Coal Sample 1, by 0.00128 for Coal Sample 2, by 0.00079 for Coal Sample 3, and by 0.00198 for Coal Sample 4 (shown in Table 2). The strains in the three directions are different. For the stable growth zone, when the gas pressure reaches about 5 MPa, the deformation of the coal samples is relatively stable, and the deformation in all three directions and volumetric strains show significant differences. For the rapid growth zone, when the gas pressure reaches the maximum value, the deformation in all

three directions and volumetric strains of the coal samples are in a rapid growth stage.

Table 2 Increase of volumetric strains for coal samples

Sample Growth zone	Coal Sample 1	Coal Sample 2	Coal Sample 3	Coal Sample 4
Slow growth zone	0.00084	0.00128	0.00079	0.00198
Stable growth zone	0.01044	0.0142	0.01628	0.02721
Rapid growth zone	0.02255	0.02652	0.02601	0.05529

From a microscopic point of view, the coal samples undergo expansion deformation after adsorbing gas. When the gas adsorption occurs, the gas molecules adhere to the surface of coal particles, resulting in a decrease in the surface tension of the coal particles. This means the attraction between the molecules on the surface of coal particles and the molecules inside is reduced and thus the distance between them is increased. Therefore, the coal mass after its gas adsorption is more likely to be deformed than before its gas adsorption (Li et al., 2018).

From a macroscopic point of view, the coal samples in this experiment are cuboids, and their length in the σ_1 direction is twice of those in the other two directions. Consequently, the adsorption deformation mainly occurs in the σ_1 direction. For the other two directions, rigid loading is adopted in the σ_2 direction with a large stress, and flexible loading is adopted in the σ_3 direction with a small stress. This indicates that the external binding force in the σ_2 direction is larger than that in the σ_3 direction, and thus deformation is prone to occur in the σ_3 direction. Therefore, the gas adsorption deformation exhibits the characteristics of $\varepsilon_1 > \varepsilon_3 > \varepsilon_2$.

In this gas adsorption experiment, gas pressure is the main factor of coal mass deformation. Gas pressure acts as both adsorption pressure and pore pressure in coal sample pore fractures. Specifically, the effect of gas adsorption plays a leading role under low gas pressure conditions while the effect of pore pressure plays a dominant role under high gas pressure conditions.

When the external stress of the coal mass is constant and the gas adsorption pressure is less than 5 MPa, the volumetric strains of the coal samples depend on the gas adsorption of the coal samples. Moreover, gas adsorption is related to gas pressure. As gas pressure increases, gas adsorption becomes stronger, and more gas is adsorbed on the pore surface of the coal samples resulting in a larger adsorption thickness. Consequently, the transfer resistance between gas molecules increases. When the gas pressure is low, the pore pressure is small, and thus it is difficult to cause the deformation of coal

mass. Therefore, the volumetric strains of the coal samples in this zone are mainly determined by gas adsorption.

When the gas adsorption pressure is greater than 5 MPa, the gas pressure exhibits the effect of pore pressure other than the effect of adsorption, and the interior of the coal samples expand outward owing to pore pressure. This indicates that the volumetric strains of the coal samples increase with the increase of the gas pressure. The larger the gas pressure, the larger the pore pressure, and the greater the volumetric strain increase. In this process, gas adsorption is reduced to a secondary position.

3.2 Mechanism of gas adsorption deformation

In this paper, the emphaticar density, Poisson's ratio, Young's modulus, bulk modulus, pore structure length, diameter and other parameters of soft coalbed methane reservoirs are adopted to model the coalbed methane reservoir medium (Yaodong et al., 2007). In this model, the soft coalbed methane reservoir and gas satisfy the following conditions: (1) the reservoir is an isotropic continuous medium, which means that the physical and mechanical properties are the same in all directions; (2) the reservoir is a saturated mixture composed of coal matrix skeleton and gas that is free and adsorbed in pore structures; (3) the coal matrix is always in a solid state, and the adsorbed gas is always in a gaseous state, and the two will not transform into each other; (4) the temperature of gas in reservoirs is kept constant during the adsorption and desorption process..

Gas in soft coalbed reservoirs mainly exists in an absorbed state, and the volume deformation of coal mass is caused by both pore pressure and gas adsorption. Schere (1986) believed that when gas-containing coal is assumed to be an isotropic elastic medium and the elasticity energy is equal to the change of surface energy, the expansion strain caused by gas adsorption is:

$$\varepsilon_v = \gamma A \rho_s \frac{f(\varphi, v_s)}{E_s} \quad (1)$$

$$f(\varphi, v_s) = \left(1 - \frac{4c\varphi(1-2v_s)}{3-5v_s} \right) \cdot \left(\frac{2(1-v_s) - c\varphi(1+v_s)}{2-3c\varphi} \right) \quad (1-1)$$

$$c = \frac{8\sqrt{2}}{3\pi} \quad ; \quad \varphi = \frac{a}{l} \quad (1-2)$$

[revised manuscript text omitted]

**Figure 13 Variation of permeability with principle stress difference for 4 kinds of coal**
**samples**

Figure 12 shows the forms of the 4 kinds of coal samples after the coal seepage experiment. In the
coal seepage experiment under true triaxial stress conditions, the coal samples are mainly affected by
shear stress, and obvious macroscopic fractures are formed on the surface of coal samples. The main
fractures have a horizontal angle of about 65° (green dotted line), and flaky shedding pits are formed
on the surface of the coal samples (yellow dotted line). Figure 13 shows the permeability variation
curve of the 4 kinds of coal samples under true triaxial stress conditions. As the maximum principal
stress difference ($\Delta\sigma$) increases under true triaxial stress conditions, the permeability of coal samples
decreases gradually in different stages, and these stages are indicated by different zones: a slow
reduction zone, a rapid reduction zone, and a steady reduction zone.

For the slow reduction zone, with the increase of $\Delta\sigma$ from 2 MPa to about 4.5 MPa, the
permeability of the 4 kinds of coal samples decreases to varying degrees. Specifically, it decreases by
$0.79 \times 10^{-15} \text{ m}^2$ for Coal Sample 1, by $0.77 \times 10^{-15} \text{ m}^2$ for Coal Sample 2, by $0.43 \times 10^{-15} \text{ m}^2$ for Coal
Sample 3, and by $0.71 \times 10^{-15} \text{ m}^2$ for Coal Sample 4, as shown in Table 4. The permeability decreases
slowly because the original pores of coal samples are continuously compressed, resulting in narrowed
the seepage channels.

For the rapid reduction zone, as $\Delta\sigma$ increases from 4.5 MPa to about 12 MPa, the permeability
continues to decrease rapidly and approximately linearly, and the coal samples undergo elastic
deformation. Now the coal samples enter the elastic stage and they are in a state of compression
deformation. If the pressure relief measures are taken, the coal samples deformed can be restored to
their original state. However, the original pores and fractures of coal samples continue to be
compressed, the permeability continues to decrease rapidly.

For the steady reduction zone, as $\Delta\sigma$ increases from about 12 MPa to 18 MPa, the coal samples are
subjected to large true triaxial stresses. This leads to a transition of coal samples from a compressed
state to an expanded state. New micro fractures are generated owing to the relative slip between the
internal particles of the coal samples, finally resulting in macro fractures. However, the closure of the
original fractures counteracts the generation of new fractures, causing the permeability of coal samples
to decrease slowly. If σ_1 continues to be increased at this time, the coal samples are destructed after

they reach their peak strength. As a consequence, the permeability of coal samples exhibits an upward trend.

Table 4 Variation of permeability difference for coal samples

Permeability	Coal sample 1	Coal sample 2	Coal sample 3	Coal sample 4
Initial permeability $\times 10^{-15} \text{ m}^2$	6.38	7.21	6.52	5.27
Slow reduction zone - permeability difference $\times 10^{-15} \text{ m}^2$	0.79	0.77	0.43	0.71
Rapid reduction zone - permeability difference $\times 10^{-15} \text{ m}^2$	3.72	4.77	4.92	3.81
Steady reduction zone - permeability difference $\times 10^{-15} \text{ m}^2$	0.36	0.33	0.16	0.03

3.4 Evolution mechanism of permeability for soft coal mass

Chikatamarla et al. (2004) found through gas adsorption experiments that the volumetric strain of coal induced by gas adsorption is proportional to the amount of gas adsorbed. According to rock mechanics, the stress and strain of coal deformation can be expressed as (Palciauskas and Domenico, 1982):

$$\sigma_{ij} = \frac{E_s}{1+\nu_s} (\varepsilon_{ij} + \frac{\nu_s}{1-2\nu_s} \varepsilon_{cv} \delta_{ij}) + \zeta p \delta_{ij} + K_s \varepsilon_v \delta_{ij} \quad (8)$$

$$K_s = \frac{E_s}{3(1-2\nu_s)} \quad (8-1)$$

[revised manuscript text omitted]

Li, X.C., Zhang, L., Li, Z.B., Jiang, Y., Nie, B.S., Zhao, Y.L., Yang, C., 2018. Creep law and model of coal under
triaxial loading at different gas pressures. *J. China Coal Soc.* 43, 473–482.

Liu, J., Chen, Z., Elsworth, D., Miao, X., Mao, X., 2011. Evolution of coal permeability from stress-controlled to
displacement-controlled swelling conditions. *Fuel* 90, 2987–2997. <https://doi.org/10.1016/j.fuel.2011.04.032>

Liu, T., Lin, B., Yang, W., 2017. Impact of matrix–fracture interactions on coal permeability: Model development
and analysis. *Fuel* 207, 522–532. <https://doi.org/10.1016/j.fuel.2017.06.125>

Liu, Y., Li, M., Yin, G., Zhang, D., Deng, B., 2018. Permeability evolution of anthracite coal considering true
triaxial stress conditions and structural anisotropy. *J. Nat. Gas Sci. Eng.* 52, 492–506.
<https://doi.org/10.1016/j.jngse.2018.02.014>

Lu, M., Connell, L., 2016. Coal failure during primary and enhanced coalbed methane production - Theory and
approximate analyses. *Int. J. Coal Geol.* 154–155, 275–285. <https://doi.org/10.1016/j.coal.2016.01.008>

Lu, S., Cheng, Y., Li, W., 2016. Model development and analysis of the evolution of coal permeability under
different boundary conditions. *J. Nat. Gas Sci. Eng.* 31, 129–138. <https://doi.org/10.1016/j.jngse.2016.02.049>

Meng, Y., Li, Z., 2017. Triaxial experiments on adsorption deformation and permeability of different sorbing gases
in anthracite coal. *J. Nat. Gas Sci. Eng.* 46, 59–70. <https://doi.org/10.1016/j.jngse.2017.07.016>

Mitra, A., Harpalani, S., Liu, S., 2012. Laboratory measurement and modeling of coal permeability with continued
methane production: Part I - Laboratory results. *Fuel* 94, 110–116. <https://doi.org/10.1016/j.fuel.2011.10.052>

Myers, A.L., 2002. Thermodynamics of adsorption in porous materials. *AIChE J.* 48, 145–160.
<https://doi.org/10.1002/aic.690480115>

Palciauskas, V. V., Domenico, P.A., 1982. Characterization of drained and undrained response of thermally loaded
repository rocks. *Water Resour. Res.* 18, 281–290. <https://doi.org/10.1029/WR018i002p00281>

PALMER I, MANSOORI J. , 1998. How Permeability Depends on Stress and Pore Pressure in Coalbeds: A New
Model. *SPE Reservoir Eval. & Eng.* 1, 539–544. <https://doi.org/10.2118/36737-MS>

Pan, Z., Connell, L.D., 2012. Modelling permeability for coal reservoirs: A review of analytical models and testing
data. *Int. J. Coal Geol.* 92, 1–44. <https://doi.org/10.1016/j.coal.2011.12.009>

Pan, Z., Connell, L.D., 2007. A theoretical model for gas adsorption-induced coal swelling. *Int. J. Coal Geol.* 69,
243–252. <https://doi.org/10.1016/j.coal.2006.04.006>

Pan, Z., Haque, A., Tan, Y., Connell, L.D., Liu, J., Sun, W., Zhou, F., 2018. Experimental study of impact of
anisotropy and heterogeneity on gas flow in coal. Part II: Permeability. *Fuel* 230, 397–409.
<https://doi.org/10.1016/j.fuel.2018.05.033>

Peng, Y., Liu, J., Pan, Z., Connell, L., Chen, Z., Fuel, H.Q., 2017, U., 2017. Impact of coal matrix strains on the
evolution of permeability. *Fuel* 189, 270–283.

Peng, Y., Liu, J., Wei, M., Pan, Z., Connell, L.D., 2014. Why coal permeability changes under free swellings: New
insights. *Int. J. Coal Geol.* 133, 35–46. <https://doi.org/10.1016/j.coal.2014.08.011>
Qian, M.G., Xu, J. I., Wang, J.C., 2018. Further on the sustainable mining of coal. *J. China Coal Soc.* 43, 1–13.
<https://doi.org/10.13225/j.cnki.jccs.2017.4400>
Sato, M., Takemura, T., Takahashi, M., 2018. Development of the permeability anisotropy of submarine
sedimentary rocks under true triaxial stresses. *Int. J. Rock Mech. Min. Sci.* 108, 118–127.
<https://doi.org/10.1016/j.ijrmms.2018.06.010>
Saurabh, S., Harpalani, S., 2018. Stress path with depletion in coalbed methane reservoirs and stress based
permeability modeling. *Int. J. Coal Geol.* 185, 12–22. <https://doi.org/10.1016/j.coal.2017.11.005>
Schere, G.W., 1986. Dilatation of Porous Glass. *J. Am. Ceram. Soc.* 69, 473–480.
<https://doi.org/10.1111/j.1151-2916.1986.tb07448.x>
Shi, J.Q., Durucan, S., 2004. Drawdown induced changes in permeability of coalbeds: A new interpretation of the
reservoir response to primary recovery. *Transp. Porous Media* 56, 1–16.
<https://doi.org/10.1023/B:TIPM.0000018398.19928.5a>
Wang, G., Li, W., Wang, Pengfei, Yang, X., Zhang, S., 2016. Deformation and gas flow characteristics of coal-like
materials under triaxial stress conditions. *Int. J. Rock Mech. Min. Sci.* 91, 72–80.
<https://doi.org/10.1016/j.ijrmms.2016.11.015>
Wang, G., Pang, Y., 2017. Surrounding rock control theory and longwall mining technology innovation. *Int. J. Coal*
*Sci. Technol.* 4, 301–309. <https://doi.org/10.1007/s40789-017-0188-8>
Wang, G., Wang, P., Guo, Y., Li, W., 2018. A Novel True Triaxial Apparatus for Testing Shear Seepage in
Gas-Solid Coupling Coal. *Geofluids* 2018, 1–9. <https://doi.org/10.1155/2018/2608435>
WANG Gang, CHENG Wei-min, GUO Heng, 2012. Study on permeability characteristics of coal body. *J. Min. Saf.*
*Eng.* 29, 735–739.
Wang, R., Huang, Z., Liu, T., Liu, T., Zhao, Y., Yang, W., Lin, B., Kong, J., 2017. Dynamic diffusion-based
multifield coupling model for gas drainage. *J. Nat. Gas Sci. Eng.* 44, 233–249.
<https://doi.org/10.1016/j.jngse.2017.04.026>
Wei, C., Liu, L., Zhu, W., Peng, Y., Liu, J., 2017. Impact of gas adsorption-induced coal damage on the evolution of
coal permeability. *Int. J. Rock Mech. Min. Sci.* 101, 89–97. <https://doi.org/10.1016/j.ijrmms.2017.11.007>
Wu, S., Zhao, W., 2005. METHANE-COAL SYSTEM. *Chinese J. Rock Mech. Eng.* 24, 1674–1678.
Xie, H.-P., Gao, F., Zhou, H.-W., Cheng, H.-M., Zhou, F.-B., 2013. On theoretical and modeling approach to
mining-enhanced permeability for simultaneous exploitation of coal and gas. *Meitan Xuebao/Journal China Coal Soc.* 38.
<https://doi.org/10.13225/j.cnki.jccs.2013.07.016>
Yaodong, J., Jie, Z., Yixin, ZHAO, Jinghong, L., Hongwei, W., 2007. Constitutive equations of coal constitutive
equations of coal. *J. China Coal Soc.* 32, 1132–1137.
Yin, G., Li, M., Wang, J.G., Xu, J., Li, W., 2015. Mechanical behavior and permeability evolution of gas infiltrated
coals during protective layer mining. *Int. J. Rock Mech. Min. Sci.* 80, 292–301.
<https://doi.org/10.1016/j.ijrmms.2015.08.022>
Zang, J., Wang, K., 2017. Gas sorption-induced coal swelling kinetics and its effects on coal permeability evolution:
Model development and analysis. *Fuel* 189, 164–177. <https://doi.org/10.1016/j.fuel.2016.10.092>
Zhu, J., Jiang, Y., Zhao, Y., 2009. Constitutive relation for gas-filled coal considering adsorption. *Yanshilixue Yu*
*Gongcheng Xuebao/Chinese J. Rock Mech. Eng.* 28.

Dear Editor,

On behalf of my co-authors, I am submitting the enclosed manuscript “Influence of Gas Migration on
Permeability of Soft Coalbed Methane Reservoirs under True Triaxial Stress Conditions” for possible
publication in ROYAL SOCIETY OPEN SCIENCE.

We certify that we have participated sufficiently in the work to take public responsibility for the
appropriateness of the experimental design and method, and the collection, analysis, and interpretation
of the data.

We have reviewed the final version of the manuscript and approve it for publication. The manuscript
has not been published in whole or in part nor is it being considered for publication elsewhere.

Yours Sincerely,

**Author:** Gang Wang ^{a, b, *}, Zhiyuan Liu ^b, Yanwei Hu ^b, Cheng Fan ^b, Wenrui Wang ^b, Jinzhou Li ^b
a Shandong University of Science and Technology, Mine Disaster Prevention and Control-Ministry of
State Key Laboratory Breeding Base, Qingdao 266590, China
b Shandong University of Science and Technology, College of Mining and Safety Engineering,
Qingdao 266590, China

We deeply appreciate your consideration of our manuscript, and we look forward to receiving
comments from the reviewers. If you have any queries, please do not hesitate to contact me at the
address below.

Thank you and best regards.

Yours sincerely,

Gang Wang

**Corresponding author:** Gang Wang

**Corresponding author at:** Room 429, College of Mining and Safety Engineering, Shandong
University of Science and Technology, 579 Qianwangang Rd, Huangdao District, Qingdao 266590,
China.

**Tel:** +86-13615327361

**E-mail:** gang.wang@sdust.edu.cn

Highlights:

(1) Under true triaxial stress conditions, the gas adsorption and seepage experiments of soft coal mass is carried out, and the obtained permeability curve exhibits 2 obvious turning points.

(2) The adsorption deformation law of soft coal body under the coupling of true triaxial stress field and gas pressure field is studied, which conforms to the established gas adsorption deformation model.

(3) The permeability model of soft coal body considering gas adsorption is established, and the evolution mechanism of soft coal body permeability under the condition of true triaxial stress is revealed.

Appendix B

Manuscript Number: RSOS-190892.R1

Title: Influence of Gas Migration on Permeability of Soft Coalbed Methane Reservoirs under True Triaxial Stress Conditions

Dear Editor and Reviewers,

Thank you for your comments on our manuscript entitled “**Influence of Gas Migration on Permeability of Soft Coalbed Methane Reservoirs under True Triaxial Stress Conditions**” (Manuscript Number: **RSOS-190892.R1**). These comments are all valuable and very helpful for revising and improving our paper, as well as the important guiding significance to our researches. We have studied the comments carefully and have made corrections, which we hope to meet with approval. Revised portions are marked in red in the manuscript. The main corrections in the paper and the responses to the reviewers’ comments are as follows.

Looking forward to hearing from you.

Sincerely yours,

Gang Wang

Reviewer: 1

Comments to the Author(s)

1. A number of experimental measurements have been undertaken on Coal samples looking at the impact of gas sorption and changes in permeability related to strain. The changes in strain and the permeability changes are then modelled using a combination of rock mechanics deformation models, a sorption model and a permeability model. The paper is well presented.

As there is only a "yes // no" option to a number of the questions I have been asked, I have been forced to choose the "no" option, where in most cases it is close to being acceptable. However there are a number of key issues which need to be clarified or revisited. I have attached a commented manuscript which will aid the authors in reviewing their paper.

Response: Thank you for your recognition and valuable comments on this article. For your question, we have explained accordingly. In order to facilitate your review, the revised part has been highlighted in red.

2. A key issue is that the authors identify a "slow", "stable" and "Rapid" sample growth zone, based on the relationship to an elastic model describing the relationship between gas pressure and sample strain. However there is no clear inclusion of the triaxial state of pressure in this model, and it appears that a formulation is used that uses a value of stress as the surface potential energy. The issue is that the division of stable to rapid is occurring when the gas pressure is significantly above two of the axial pressures. The model is not able to account for the triaxial stress, and thus I believe this is significantly influencing the results and the interpretation. i.e. the model used is not really valid for this type of interpretation. I would suggest revisiting the results with this in mind, as the experimental results are certainly good quality. Also the modelling results show good correlations. However process understanding is key.

Response: First of all, thank you very much for your advice. We strongly agree with the your comment of "process understanding is the key". The innovation of this paper is to combine the true triaxial coal body adsorption and seepage test with the theoretical model. In fact, the theoretical model of this paper considers the true triaxial stress factor, but the transformation of stress into true triaxial strain is indicated in the paper. As shown in the paper, the adsorption volumetric strain is calculated by the strain in the three stress directions. If the volumetric strain is reconverted into the three-direction

strain, the adsorption strain model is complicated. For the sake of brevity, the author has transformed the true triaxial stress.

The starting point of the model proposed by the experts is to establish the surface potential energy, and we also draw on the research results of the predecessors. For example, Vandamme et al. (2010) established a model between stress, strain, pore pressure and free energy from the perspective of energy balance. Kowalczyk et al. (2008) studied the relationship between adsorption stress and adsorption deformation from the perspective of thermodynamics. Moore (2012) reviewed the production and development of coalbed methane, pointed out the importance of coalbed methane as a clean fuel, and reported the adsorption characteristics of coal as a key factor in the study of coalbed methane utilization. With the increase of the adsorption pressure, the adsorption capacity of coalbed methane is gradually enhanced. Ottiger et al. (2008) studied the adsorption capacity of methane and carbon dioxide in coal. As the adsorption pressure increases, the amount of gas adsorbed on coal increases. Li et al. (2010) studied the adsorption characteristics of methane under high adsorption pressure conditions. Pini et al. (2010) studied the adsorption characteristics of methane, carbon dioxide, nitrogen, etc., and established an adsorption model from the perspective of thermal energy. Battistutta et al. (2010) also studied the adsorption energy of nitrogen, carbon dioxide and methane, established an adsorption model by combining the adsorption amount and temperature, and pointed out that as the adsorption time increases, the adsorption pressure of the gas becomes smaller and smaller. Karacan (2007) studied the volumetric strain caused by gas adsorption by means of scanning electron microscopy. With Monte Carlo and molecular dynamics, Zhang et al. (2014) simulated the adsorption of methane under different adsorption pressures and different coal body moisture contents. Busch and Gensterblum (2011) summarized the research status of coalbed methane adsorption deformation, and pointed out that the adsorption deformation increases with the increase of the coal machine rank, especially when the adsorption pressure is lower than 10 MPa. The detailed research status is shown in Table 1.

Table1 Status of research of establishing adsorption deformation model based on energy and dynamics

No.	Title	Research content
1	Adsorption and Strain: The CO ₂ -Induced Swelling of Coal	As long as the microstructure is well enough characterized, the volumetric strain induced by surface energy effects can therefore precisely be calculated theoretically. This calculation will be possible, however, only if the surface stress which prevails at the interface between the solid matrix and the pore fluid is known. $\underline{\underline{\varepsilon_a}} = -\underline{\underline{\sigma^s}} \frac{8\pi R^2 b}{\Omega_0 \phi} \underline{\underline{S}}^{hom} : \underline{\underline{1}} = -\underline{\underline{\sigma^s}} \frac{8\pi R^2 b}{\Omega_0 \phi} \frac{1}{9K} \underline{\underline{1}}$
2	Adsorption-Induced Deformation of Microporous Carbons: Pore Size Distribution Effect	An advantage of Eq. (14) is avoiding the differentiation of isotherms with respect to the pore size; however, the fluid density must be calculated quite accurately. In the current paper we determined adsorption stress from Eq. (13) by using the grand Canonical Monte Carlo simulation method. $\sigma_s(H, \mu) = k_B T \frac{\partial N(\mu_r)}{\partial H} + \int_{\mu_r}^{\mu} \frac{\partial N}{\partial H} d\mu \dots\dots\dots(13)$ $\sigma_s(H, p) = - \int \rho(z) \frac{\partial V_{sf}(z, H)}{\partial H} dz \dots\dots\dots(14)$
3	Coalbed methane: A review	Example of two types of adsorption isotherm curves: high rank coal (from the Pocahontas coal seam in the Appalachian basin) and low rank coal (from the Powder River basin). Note that the high rank coal has an initial steep slope and then flattens whereas the low rank coal tends to have a steady slope. 4	Combined Monte Carlo and molecular dynamics simulation of methane adsorption on dry and moist coal	Adsorbed and bulk densities of CH₄ on dry and moist coal as a function of pressure at 308 K. 
3. Likewise the permeability model appears to be assuming no axial strain in the s₂ or s₃ direction, however the experimental results show strain in these directions. I have made a number

of more detailed comments on the way through the manuscript.

Response: Here we put forward our explanation for Comment 3 and Comment 4 together. Equation 8 and 9 in this paper are descriptions of stress in three directions. In order to obtain the initial true triaxial stress state of the coal body, the initial true triaxial stress is calculated by the method of Zhenxiong and Peitao (2018). Combined with the test process of this true triaxial seepage test, the true triaxial stress is first generated, then the maximum principal stress is changed, and the variation law of permeability during the dynamic evolution of true triaxial stress is studied. It is not a model in which the permeability changes as the three stresses change. Therefore, the established permeability model is consistent with the true triaxial seepage process. In the meantime, the influence of adsorption deformation on permeability is considered.

Connell et al. (2010) reviewed the evolution of the permeability model under triaxial stress-strain conditions. L. D. Connell (2009) established a coupled numerical model and used the model to study the applicability of the geomechanical assumptions for coal seam gas drainage. Chen et al. (2015) studied the effects of effective stress and reservoir pressure on shale permeability, and established a permeability model related to two factors. Wu et al. (2010) pointed out that the permeability and porosity vary with the effective stress, and the adsorption-induced strain forms the overall stress and increases the effective stress. H. Zhang et al. (2008) studied the influence of coal adsorption deformation on coal porosity and permeability. The finite element model was used to quantitatively analyze the coal permeability change rate, gas flow and deformation. Siriwardane et al. (2009) studied the coal permeability of carbon dioxide under different confining pressures, pointing out that the coal permeability gradually decreased with the increase of the confining pressure. Liu et al. (2011) studied the applicability of the uniaxial strain permeability model and the multiaxial strain permeability model, pointing out that the former was more in line with the overall behavior of coalbed methane reservoirs under typical conditions, while the latter is more suitable for the permeability law under laboratory conditions. The detailed research status is shown in Table 1.

Table 2 Status of research on permeability evolution models

No.	Title	Research content
1	An analytical coal permeability model for tri-axial strain and stress	In this section the theoretical basis for the models developed in this paper is presented. In the next section the model derivations are presented for laboratory testing with tri-axial deformation and cylindrical geometry of

	conditions	core samples. $k = k_0 \left(\frac{\phi}{\phi_0} \right)^3 \quad k = k_0 \left\{ 1 - \frac{1}{\phi_0} \left[\frac{1}{K} (\tilde{p}_c - \tilde{p}_p) + (\tilde{\varepsilon}_b^{(S)} - \tilde{\varepsilon}_m^{(S)}) \right] \right\}^3 \quad (1); \quad (2);$ $k = k_0 \exp \left\{ -3 \left[C_{pc}^{(M)} \left[\frac{2(1+\nu)}{3} \tilde{p}_r^* - \left(\frac{\alpha E}{9K} + 1 \right) \tilde{p}_p - \frac{E}{9} \tilde{\varepsilon}_b^{(S)} \right] - (1-\gamma) \tilde{\varepsilon}_b^{(S)} \right] \right\} \quad (3)$
2	Coupled flow and geomechanical processes during gas production from coal seams	Shi and Durucan (2004) derived the relationship between the coal permeability and the effective stress using the following incremental stress-strain relationships, Eq. (3), for each normal effective stress component. $\begin{aligned} \bar{\sigma}_{xx}^e &= E \left[\frac{1}{1+\nu} \bar{\varepsilon}_{xx} + \frac{\nu}{(1+\nu)(1-2\nu)} \bar{\varepsilon} + \frac{1}{3(1-2\nu)} \bar{\varepsilon}^3 \right] \\ \bar{\sigma}_{yy}^e &= E \left[\frac{1}{1+\nu} \bar{\varepsilon}_{yy} + \frac{\nu}{(1+\nu)(1-2\nu)} \bar{\varepsilon} + \frac{1}{3(1-2\nu)} \bar{\varepsilon}^3 \right] \\ \bar{\sigma}_{zz}^e &= E \left[\frac{1}{1+\nu} \bar{\varepsilon}_{zz} + \frac{\nu}{(1+\nu)(1-2\nu)} \bar{\varepsilon} + \frac{1}{3(1-2\nu)} \bar{\varepsilon}^3 \right] \dots\dots\dots (3) \end{aligned}$ The relationship between the coal permeability for each axis and the effective stress can be expressed by Eq. (8). $k_i = k_{i0} e^{-3C_{b0,i}(\sigma_{ij}^e - \sigma_{j0}^e)} \dots\dots\dots (8)$
3	Dual poroelastic response of a coal seam to CO ₂ injection	The relation between matrix permeability ratio and matrix pore pressure at a specific point is shown in Fig.6. the permeability ratio increases with an increase in the matrix pore pressure as expected with the effective stress dependency. In these situations, the effective stress effect and the sorption effect are competing: an increase in the matrix pressure..... For these particular conditions the resultant effect is a monotonic decrease in permeability with increasing pressure as the effects of sorption-induced swelling dominate. Fig. 6. The relation between matrix permeability ratio and matrix pore pressure at the specific point of $x = 2$ and $y = 2$, adjacent to the wellbore.
4	Combined Monte Carlo and molecular dynamics simulation of methane adsorption on dry and moist coal	Assuming thermal expansion/contraction and matrix swelling/shrinkage are isotropic, the stress-strain relationships for a non-isothermal coalbed may be written as (negative in compression). $\Delta \varepsilon_{ij} = \frac{1}{2G} \Delta \sigma_{ij} - \left(\frac{1}{6G} - \frac{1}{9K} \right) \Delta \sigma_{kk} \delta_{ij} + \frac{\alpha}{3K} \Delta p \delta_{ij} + \frac{\Delta \varepsilon_s}{3} \delta_{ij} + \frac{\alpha_T}{3} \Delta T \delta_{ij}$ The exponential relation was used for the permeability calculation: $\frac{k}{k_0} = \exp \left(-\frac{3\Delta \sigma_e^h}{E_f} \right) = \exp \left[-\frac{3}{E_f} \left(-\frac{\nu}{1-\nu} \Delta p + \frac{E}{1-\nu} \gamma \Delta S \right) \right]$ The cubic relation between porosity and permeability was used for this derivation, as shown below:

		$\frac{k}{k_0} = \left[1 + \frac{c_m}{\phi_0}(p-p_0) + \frac{\varepsilon_L}{\phi_0} \left(\frac{K}{M} - 1 \right) \left(\frac{p}{P_L + p} - \frac{p_0}{P_L + p_0} \right) \right]^3$
--	--	---

4. The permeability model developed appears principally to be a 1D s1 approximation, which may explain the slightly less accurate fitting of the experimental results.

Response: Thank you for your valuable comments. Please see the previous explanation.

5. The results where gas pressure above a principal axis stress has been allowed need to be revisited, as there is then no real control on strain.

Response: Thanks for your advice. Our previous ideas coincide with your thoughts. We considered the gas sealing problem at that time, that is, as the adsorption deformation and the adsorption pressure increase, especially when the gas pressure exceeds the maximum principal stress, how we seal the gas so that the gas does not leak. The reason why we did not change the axial strain is we have two layers of heat-shrink tubing around the coal sample. This is found to be effective from the gas adsorption test results.

6. Although the experimental work is truly triaxial, it appears the modelling is not. This is fine as long as the limitations and best approximations are included and an understanding of what the limiting factors may be.

Response: Thank you very much for your recognition of our research work. The model in this paper may not be seen as a true triaxial model from the expression. The reason why we did this is explained in the article. our subsequent research work is to establish a more accurate true triaxial model. The true triaxial test of this paper lays an important experimental foundation for our model derivation in future.

7. The title needs rethinking a bit as it doesn't really reflect the contents, currently emphasises the flow aspect of gas, but the work is all about gas pressures and stress conditions.

Response: Thanks for your valuable suggestion. We attach great importance to this. The title for this paper is changed. The gas migration visualization and the true triaxial model in the coal are important tasks that we will carry out later. For example, the former may be achieved by using nuclear magnetic resonance technology and CT scanning technology. However, in this paper, the effects of gas pressure and true triaxial stress environment on gas migration in coal are mainly studied.

Reviewer: 2

Comments to the Author(s)

1. The author designed a gas adsorption and seepage test for soft coal under true triaxial stress conditions for soft coal. The stress conditions are in line with coal mining. The true three-dimensional stress state in the process, the research object and the research method are very innovative. Based on the true triaxial test, the author established a theoretical model of coal gas adsorption deformation and a model of seepage evolution. The results of the two models are similar to those exhibited by the true triaxial test. It is the verification of the accuracy of the true triaxial test and the verification of the applicability of the adsorption and seepage models. The analysis is clear and logical. Therefore, the research results of this paper are very valuable, and can provide experimental and theoretical guidance for coalbed methane extraction, and it is recommended to accept after minor revision.

Response: Thank you for your recognition and valuable comments on this article. For your question, we have explained accordingly. In order to facilitate your review, the revised part has been highlighted in red.

2. The true triaxial seepage test system of this paper is independently developed by the author. I am interested in this. I propose a design pattern of “two rigid and one soft, three-way independent loading”. Why should we design this way? What are the advantages? Can the author explain?

Response: Thank you very much for your recognition of our research work. It is mentioned in this paper to overcome the jamming problem in the three-way rigid loading process. Because in the three-direction rigid loading, the rigid indenter will generate a large frictional force against the surface of the coal sample, and the coal body cannot be released smoothly under pressure deformation, which will easily cause the indenter to get stuck and interrupt the test.

3. The manuscript text on page 2, line 34 is not used properly. The author says “the permeation law of shale”. This is not accurate. It is recommended that the author change to “Seepage Regulation of shale”.

Response: Thank you very much for your suggestion. This phrase has been revised, and highlighted in red.

4. The adsorption and percolation tests in this paper all require high sealing performance of the test equipment, and the author does not explain how the coal sample is sealed. Is it because of the length of the article? If yes, please explain here; if not, please add the reason to the body.

Response: Thank you very much for your suggestion. In fact, one of the innovations of the true triaxial gas-solid coupling coal seepage test system is the realization of gas sealing, which has been discussed in our previous article. Please refer to Wang et al. (2018).

5. The author of Figure 1 is very beautiful, but there are three other signs in the lower left corner. If it is related to the device, please mark it. If not, please delete it.

Response: Thank you very much for your suggestion. This has been revised.

6. From the author's industrial analysis of coal samples, it is known that raw coal is relatively soft and not easy to form. The author solved this problem by using coal briquettes. This method is feasible and worth promoting. But does the author consider the consistency of other mechanical properties of briquette with raw coal?

Response: Thank you for your recognition and valuable comments on the article. The authors tested the mechanical properties of both raw coal samples (listed in the paper) and finally selected briquettes samples for this test.

7. In Figure 3, the first three coal samples are the same color, and the fourth coal sample has a high exposure. Can the author be replaced by a uniform one? Is there no uniform exposure after the end of the test?

Response: Thank you very much for your suggestion. The photos were taken with a mobile phone camera after the experiment without photo retouching. We apologize for not being able to replace them.

8. It can be seen from Fig. 4-7 that when the gas adsorption pressure is lower than 1 MPa, the deformation of the coal sample in three directions is inconsistent, and the variation law is not uniform. According to the experience of reviewing experts, this stage is indeed complicated. The deformation of coal samples is dominated by stress or gas adsorption, and even the creep characteristics of coal samples. There is no unified understanding. How does the author understand the mechanism of action of gas adsorption at lower pressures?

Response: Thank you for your recognition and valuable comments on this article. The authors believe that the effect of gas adsorption and external stress on coal sample deformation is related to the

coal sample body at low pressures. If the coal sample has strong adsorption, the gas adsorption should be dominant at low pressures.

9. In the manuscript, on page 10, line 22, the author points out that “the coal mass after its gas adsorption is more likely to be deformed than before its gas adsorption”. This view is correct, but for the language to be more concise, it is recommended that the author “before” Delete later.

Response: Thank you very much for your suggestion. This has been revised, and highlighted in red.

10. Similarly, on page 10, line 52, the author wants to express that the higher the adsorption pressure, the higher the amount of gas adsorbed? The authors say "and more gas is adsorbed on the pore surface of the coal samples resulting in a larger adsorption thickness." The suggestion is changed to "and more gas is adsorbed on the pore surface of the coal samples, resulting in a larger adsorption thickness."

Response: Thank you very much for your suggestion. This has been revised, and highlighted in red.

11. On page 11, page 28 of the manuscript, the three stress directions are mistake, and there are two periods at the end of the paragraph. I hope the author can check all the writing formats in the text.

Response: Thank you very much for your suggestion. This has been revised, and highlighted in red.

12. What does "model:WSXFBX1(USER)" in Figure 8 mean? The same as Figure 9-11 below, is the fitted model? Is it different from the theoretical model derived by the author?

Response: Thanks. "model:WSXFBX1(USER)" is named by the author himself, which is consistent with all the parameters of the model being derived.

13. On page 15, line 15 of the manuscript, the unit of gas flow is error.

Response: Thank you very much for your suggestion. This has been revised, and highlighted in red.

14. There is a problem with the format of Table 3.

Response: Thank you very much for your suggestion. This has been revised, and highlighted in red.

15. The format of the headings of Figures 10 and 11 is not uniform.

Response: Thank you very much for your suggestion. This has been revised, and highlighted in red.

16. From the true triaxial adsorption test in Figure 8-11, two turning points can be clearly observed, but the theoretical model curve does not show a turning point. The reason is also explained in detail in the article, but the author can correct the model parameters and make the theory Is the model better matched to the test results?

Response: Thank you very much for your suggestion. This is the law exhibited by the true triaxial test, and the theoretical model does not have these two turning points. In our subsequent research, we will focus on establishing true triaxial adsorption and seepage models. However, this is still in progress.

17. The unit of density in Table 4 is not written correctly.

Response: Thank you very much for your suggestion. This has been revised, and highlighted in red.

18. Is the true triaxial seepage test in this paper based on the previous adsorption test? Or is the two tests separate? From the author's point of view, the two tests are carried out separately, so that the coal sample failure morphology after the true triaxial gas adsorption can be obtained, and the author is established to conduct the continuity test. If it is a continuous test, is the form of coal sample adsorption deformation obtained in a separate study of gas adsorption?

Response: Thank you very much for your suggestion. True triaxial adsorption and percolation tests are performed separately. The adsorption deformation parameters were obtained by the true triaxial adsorption test, which provided parameter support for the true triaxial seepage test and model.

19. Like the true triaxial adsorption test, the seepage test also showed a turning point, which is similar to the theoretical model of seepage. Can the author also make parameter corrections to keep the two closer?

Response: Thank you for your recognition and valuable comments on this article. In our subsequent research, we will focus on establishing true triaxial adsorption and seepage models. However, this is still in progress.

References

- Battistutta, Elisa, Patrick van Hemert, Marcin Lutynski, Hans Bruining, and Karl Heinz Wolf. 2010. "Swelling and Sorption Experiments on Methane, Nitrogen and Carbon Dioxide on Dry Selar Cornish Coal." *International Journal of Coal Geology* 84 (1): 39–48.
<https://doi.org/10.1016/j.coal.2010.08.002>.
- Busch, Andreas, and Yves Gensterblum. 2011. "CBM and CO₂-ECBM Related Sorption Processes in Coal: A Review." *International Journal of Coal Geology* 87 (2): 49–71.
<https://doi.org/10.1016/j.coal.2011.04.011>.
- Chen, Dong, Zhejun Pan, and Zihui Ye. 2015. "Dependence of Gas Shale Fracture Permeability on Effective Stress and Reservoir Pressure: Model Match and Insights." *Fuel* 139: 383–92. <https://doi.org/10.1016/j.fuel.2014.09.018>.
- Connell, L. D. 2009. "Coupled Flow and Geomechanical Processes during Gas Production from Coal Seams." *International Journal of Coal Geology* 79 (1–2): 18–28.
<https://doi.org/10.1016/j.coal.2009.03.008>.
- Connell, Luke D., Meng Lu, and Zhejun Pan. 2010. "An Analytical Coal Permeability Model for Tri-Axial Strain and Stress Conditions." *International Journal of Coal Geology* 84 (2): 103–14. <https://doi.org/10.1016/j.coal.2010.08.011>.
- Karacan, C. Özgen. 2007. "Swelling-Induced Volumetric Strains Internal to a Stressed Coal Associated with CO₂ Sorption." *International Journal of Coal Geology* 72 (3–4): 209–20.
<https://doi.org/10.1016/j.coal.2007.01.003>.
- Kowalczyk, Piotr, Alina Ciach, and Alexander V Neimark. 2008. "Adsorption-Induced Deformation of Microporous Carbons: Pore Size Distribution Effect," no. 32: 6603–8.
- Li, Dongyong, Qinfu Liu, Philipp Weniger, Yves Gensterblum, Andreas Busch, and Bernhard M. Krooss. 2010. "High-Pressure Sorption Isotherms and Sorption Kinetics of CH₄ and CO₂ on Coals." *Fuel* 89 (3): 569–80. <https://doi.org/10.1016/j.fuel.2009.06.008>.
- Liu, Jishan, Zhongwei Chen, Derek Elsworth, Hongyan Qu, and Dong Chen. 2011. "Interactions of Multiple Processes during CBM Extraction: A Critical Review." *International Journal of Coal Geology* 87 (3–4): 175–89. <https://doi.org/10.1016/j.coal.2011.06.004>.
- Moore, Tim A. 2012. "Coalbed Methane: A Review." *International Journal of Coal Geology* 101: 36–81. <https://doi.org/10.1016/j.coal.2012.05.011>.

Ottiger, Stefan, Ronny Pini, Giuseppe Storti, and Marco Mazzotti. 2008. "Competitive Adsorption Equilibria of CO₂ and CH₄ on a Dry Coal." *Adsorption* 14 (4–5): 539–56. <https://doi.org/10.1007/s10450-008-9114-0>.

Pini, Ronny, Stefan Ottiger, Luigi Burlini, Giuseppe Storti, and Marco Mazzotti. 2010. "Sorption of Carbon Dioxide, Methane and Nitrogen in Dry Coals at High Pressure and Moderate Temperature." *International Journal of Greenhouse Gas Control* 4 (1): 90–101. <https://doi.org/10.1016/j.ijggc.2009.10.019>.

Siriwardane, Hema, Igor Haljasmaa, Robert McLendon, Gino Irdi, Yee Soong, and Grant Bromhal. 2009. "Influence of Carbon Dioxide on Coal Permeability Determined by Pressure Transient Methods." *International Journal of Coal Geology* 77 (1–2): 109–18. <https://doi.org/10.1016/j.coal.2008.08.006>.

Vandamme, M., L. Brochard, B. Lecampion, and O. Coussy. 2010. "Adsorption and Strain: The CO₂-Induced Swelling of Coal." *Journal of the Mechanics and Physics of Solids* 58 (10): 1489–1505. <https://doi.org/10.1016/j.jmps.2010.07.014>.

Wang, Gang, Pengfei Wang, Yangyang Guo, and Wenxin Li. 2018. "A Novel True Triaxial Apparatus for Testing Shear Seepage in Gas-Solid Coupling Coal." *Geofluids* 2018.

Wu, Yu, Jishan Liu, Derek Elsworth, Zhongwei Chen, Luke Connell, and Zhejun Pan. 2010. "Dual Poroelastic Response of a Coal Seam to CO₂ Injection." *International Journal of Greenhouse Gas Control* 4 (4): 668–78. <https://doi.org/10.1016/j.ijggc.2010.02.004>.

Zhang, Hongbin, Jishan Liu, and D. Elsworth. 2008. "How Sorption-Induced Matrix Deformation Affects Gas Flow in Coal Seams: A New FE Model." *International Journal of Rock Mechanics and Mining Sciences* 45 (8): 1226–36. <https://doi.org/10.1016/j.ijrmms.2007.11.007>.

Zhang, Junfang, M. B. Clennell, D. N. Dewhurst, and Keyu Liu. 2014. "Combined Monte Carlo and Molecular Dynamics Simulation of Methane Adsorption on Dry and Moist Coal." *Fuel* 122: 186–97. <https://doi.org/10.1016/j.fuel.2014.01.006>.

Zhenxiong, YAN, , and Peitao, WANG. 2018. "Insight into In-Situ Stress Calculation Applied in Hollow Inclusion Measurement." *Chinese Journal of Rock Mechanics and Engineering* 39 (2): 715–21.